



# Effectiveness of short term air quality emission controls: A high-resolution model study of Beijing during the APEC period

Tabish Umar Ansari[1], Oliver Wild[1], Jie Li[2], Ting Yang[2], Weiqi Xu[2,3], Yele Sun[2,3,4], and Zifa Wang[2]

[1]Lancaster Environment Centre, Lancaster University, UK
[2]State Key Laboratory of Atmospheric Boundary Layer Physics and Atmospheric Chemistry, Institute of Atmospheric Physics, Chinese Academy of Sciences, Beijing, China
[3]College of Earth Sciences, University of Chinese Academy of Sciences, Beijing, China
[4]Center for Excellence in Regional Atmospheric Environment, Institute of Urban Environment, Chinese Academy of Sciences, Xiamen, China

*Correspondence to:* Tabish Umar Ansari (t.ansari@lancaster.ac.uk)

**Abstract.** We explore the impacts of emission controls on haze events in Beijing in October–November 2014 using high resolution WRF-Chem simulations. The model reproduces surface temperature and relative humidity profiles over the period well and captures the observed variations in key atmospheric pollutants. We highlight the sensitivity of simulated pollutant levels to meteorological variables and model resolution, and in particular to treatment of turbulent mixing in the planetary

boundary layer. We note that simulating particle composition in the region remains a challenge, and we overpredict $NH_4$ and $NO_3$ at the expense of $SO_4$. We find that the emission controls implemented for the APEC Summit period made a relatively small contribution to improved air quality (20–26%), highlighting the important role played by favourable meteorological conditions over this period. We demonstrate that the same controls applied under less favourable meteorological conditions would have been insufficient to reduce pollutant levels to meet the required standards. Continued application of these controls

over the 6-week period considered would only have reduced the number of haze days where daily-mean fine particulate matter exceeds $75\,\mu g\,m^{-3}$ from 15 to 13 days. Our study highlights the limitations of current emission controls and the need for more stringent measures over a wider region during meteorologically stagnant weather.

## 1 Introduction

Air pollution poses serious health risks to urban residents and is one of the most important environmental problems facing cities around the world (Liang et al., 2017). Fine particulate matter with a diameter less than $2.5\,\mu m$ ($PM_{2.5}$) is a major air pollutant that often exceeds safe limits during haze episodes which are a common occurrence in many developing megacities over the past decade. It is estimated that a $10\,\mu g\,m^{-3}$ decrease in $PM_{2.5}$ concentration is related to an increase in mean life expectancy of as much as 0.6 years (Pope et al., 2009). $PM_{2.5}$ also reduces visibility and has important impacts on regional

climate (Westervelt et al., 2016). Beijing is the capital, political and cultural center of China and is among the most polluted





cities in the country (Batterman et al., 2016). The population of Beijing municipality increased from 14.2 million in 2002 to 21.2 million in 2013 (Ma et al., 2014) and this has been accompanied by an increase in anthropogenic emissions across the region. High $PM_{2.5}$ concentrations are frequently reported in city clusters in the Beijing-Tianjin-Hebei, Yangtze River Delta, and Pearl River Delta regions in China. Haze episodes are particularly common during winter months and have attracted

substantial scientific attention (Gao et al., 2017). Independent observational (Gao et al., 2016a; Zhong et al., 2018; Shang et al., 2018; Chen et al., 2015b; Sun et al., 2016a) modelling (Matsui et al., 2009; Kajino et al., 2017; Gao et al., 2015a; Chen et al., 2016a) and long-term data analysis studies (Chen et al., 2016b; Liu et al., 2016b; Chen et al., 2015a; Yan et al., 2018) have investigated the sources, evolution and fate of $PM_{2.5}$ in Beijing, but many uncertainties remain, and improved understanding is required in order to inform sound, evidence-based emission control policies. Strict short-term emission controls have been

applied effectively to improve air quality in Beijing during the Beijing Olympics in 2008 (Gao et al., 2011; Yang et al., 2011) and more recently for major events such as the Asia-Pacific Economic Cooperation (APEC) summit in November 2014 (Li et al., 2017b; Wang et al., 2016b) and the Victory Parade in 2015 (Liang et al., 2017; Liu et al., 2016a; Zhao et al., 2017). Real-world emission controls provide an ideal opportunity to test current scientific understanding of the sources and processing of air pollution as represented in models in a robust way. With improved confidence in model performance over a focus region

we can explore the impact of alternative control options to aid formulation of more effective policies for emission reduction.

A number of previous studies have investigated the effect of emission controls during the APEC period in November 2014 using surface observations (Sun et al., 2016b; Xu et al., 2015; Wang et al., 2016b; Li et al., 2017b; Zhou et al., 2017) and atmospheric chemical transport models (Zhang et al., 2016; Guo et al., 2016; Wang et al., 2017; Gao et al., 2017) and have found that $PM_{2.5}$ concentrations were much lower than during the preceding weeks. Many of these studies have attributed this

improved air quality largely to the emission controls that were applied without thoroughly evaluating the role of meteorological variations. Comparison with observations in preceding weeks or over similar time periods in earlier years does not adequately account for the role of meteorology in governing haze episodes. Model studies with and without emission controls are insufficient to evaluate the contribution of meteorological processes if they focus on the control period alone, without evaluating the model performance outside the control period. Gao et al. (2017) found that the emission controls reduced $PM_{2.5}$ levels

by about $18\,\mu g\,m^{-3}$ during APEC with about half the reduction due to emission controls in surrounding districts outside Beijing. However, the study involved coarse resolution (27 km) model simulations which may be insufficient to capture regional and city-level atmospheric events well, and lacked component level analysis of aerosols. Other studies have noted the role of meteorology during the period but have not quantified it, attributing the benefits mostly to emission controls.

In this study we investigate the effectiveness of short-term emission controls and how meteorological processes influence

this, using the APEC period as an example. We use a nested version of the Weather Research and Forecasting model with Chemistry (WRF-Chem) over China with a specific focus on the Beijing-Tianjin-Hebei region. WRF-Chem has been used successfully at coarser resolution in previous studies investigating haze formation over Beijing (Matsui et al., 2009; Tie et al., 2014; Zhang et al., 2015; Chen et al., 2016a). We describe the model setup, emissions and observations in Section 2. In Section 3 we present a thorough meteorological and chemical evaluation of the model simulations against surface observations

and tower measurements, including aerosol composition, and we assess the strengths and weaknesses of the model. We also test



different meteorological inputs to drive the model. We present sensitivity studies to model resolution, uncertainties in ammonia emissions and boundary layer processes in Section 4. In Section 5 we investigate the impact of emissions controls over the APEC period and compare these with the same controls over a period two weeks earlier to demonstrate the important role of meteorological conditions in governing their effectiveness.

## 2   Model configuration and the APEC period

We use the WRF-Chem model (Grell et al., 2005; Fast et al., 2006) version 3.7.1 to simulate the meteorology and air quality over Northern China. Previous studies have shown that WRF-Chem is capable of reproducing air quality in China relatively well (Gao et al., 2015a, 2016b; Guo et al., 2016; Chen et al., 2016a). We use the Carbon Bond Mechanism version-Z (CBMZ) chemistry scheme coupled with the Model of Simulating Aerosol Interactions and Chemistry (MOSAIC) aerosol module (Zaveri et al., 2008). CBMZ explicitly treats 67 species with 164 gas-phase, heterogeneous and aqueous reactions, and provides a suitable compromise between chemical complexity and computational efficiency. MOSAIC uses a sectional approach with eight aerosol size bins and treats the key aerosol species, including sulfate, nitrate, chloride, ammonium, sodium, black carbon (BC), primary organic mass, liquid water and other inorganic mass. Secondary organic aerosol (SOA) formation is not included in the chemical mechanism used here. While SOA may contribute as much as 17–23% to aerosol composition in the October–November period investigated here, it does not respond strongly to emission controls (Sun et al., 2016b). In contrast secondary inorganic aerosols contribute up to 62% by mass of total fine particulate matter over the North China Plain, and show a significant decrease in response to emission controls (Sun et al., 2016b), and our emphasis is largely on these components here.

Further details of the model configuration used in this study are given in Table 1. We perform two-way coupled simulations with three nested domains that include China as the parent domain (D01) at 27 km horizontal resolution, Northern China as a nest (D02) at 9 km resolution and the North China Plain as an innermost nest (D03) at 3 km resolution, as shown in Fig. 1. The model is nudged to meteorological reanalysis data above the boundary layer every six hours for winds, temperature and moisture to permit direct comparison of the simulations with observed pollutant concentrations under comparable conditions.

We use anthropogenic emissions from the Multi-resolution Emission Inventory for China (MEIC) for the year 2010 (Li et al., 2017c). This provides emissions of major air pollutants including $NO_x$, CO, NMVOC, $SO_2$, $NH_3$, $PM_{2.5}$, $PM_{10}$, BC and OC from five major emission sectors that include residential, traffic, industry, power and agricultural sources, and has been used in a number of previous modelling studies (Li et al., 2015; Gao et al., 2015a; Zhang et al., 2015; Chen et al., 2015a, 2016a). Emissions were provided at the native resolution of each domain, i.e., at 27 km, 9 km and 3 km. We impose a vertical profile for these emissions over the lowest eight model levels to account for the effective source height distribution for each sector based on the distribution used for EMEP emissions (Bieser et al., 2011; Mailler et al., 2013), and impose a diurnal cycle for each source sector in the MEIC inventory. $SO_2$ emissions over the Beijing-Tianjin-Hebei region were reduced by 50% to account for strong emission reductions between 2010 and our focus year of 2014 (Zheng et al., 2018). We assume that 6% by mass of $SO_2$ is emitted as primary $SO_4$ to account for the discrepancy between high observed concentrations of $SO_4$ and low secondary production in the model (Gao et al., 2015a; Chen et al., 2016a; Li et al., 2017a). Biogenic emissions are based on





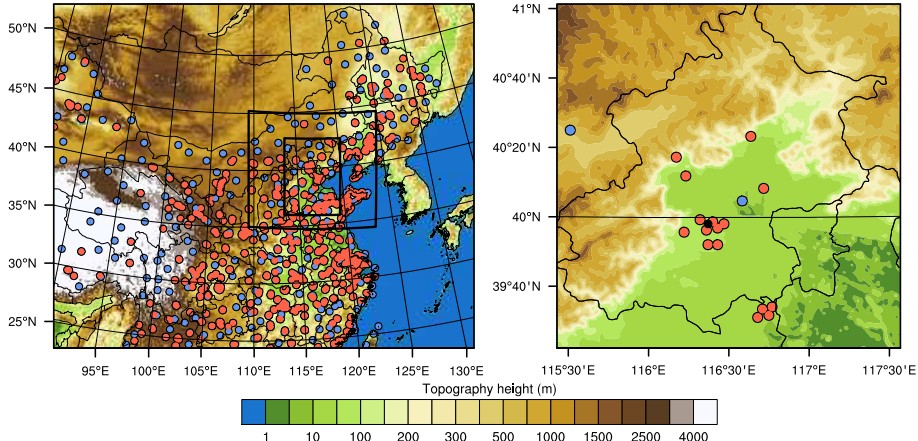

**Figure 1.** Map of topography over the model domain (left) showing nests over Northern China and the North China Plain, and map of Beijing municipality (right) showing the location of IAP (black) and measurement stations for meteorology (blue) and air quality (red).

**Table 1.** Model configuration used in this study

| Configuration | Description |
|---|---|
| Horizontal resolution | 27 km, 9 km, 3 km (3 domains) |
| Vertical levels | 31 with model top at 50 hPa |
| Aerosol scheme | MOSAIC (8 bins) (Zaveri et al., 2008) |
| Photolysis scheme | Fast-J photolysis (Wild et al., 2000) |
| Gas-phase chemistry | CBMZ (Zaveri and Peters, 1999) |
| Cumulus parameterization | Grell 3-D scheme |
| Shortwave radiation | RRTMG shortwave scheme (Clough et al., 2005) |
| Longwave radiation | RRTMG longwave scheme (Mlawer et al., 1997) |
| Cloud Microphysics | Lin scheme (Lin et al., 1983) |
| Land surface scheme | NOAH LSM (Chen and Dudhia, 2001) |
| Land-use data | MODIS 20 category at 30 arcseconds |
| Surface layer scheme | Monin-Obukhov scheme (Monin and Obukhov, 1954) |
| Boundary layer scheme | YSU (Hong et al., 2006) |
| Meteorological conditions | ECMWF 6-hourly data |
| Chemical boundary conditions | MOZART (Emmons et al., 2010) |

the Model of Emissions of Gases and Aerosols from Nature (MEGAN, Guenther et al., 2012). These are calculated online in the model based on canopy and emission factors and factors for leaf age, soil moisture, leaf area index, light dependence and



temperature responses. Hourly fire emissions are included from the Fire Emissions Inventory from NCAR (FINN, Wiedinmyer et al., 2011) to represent biomass burning, although this is not a major source in the region at this time of year.

To evaluate the model, meteorological observations were obtained from the National Climate Data Center (NCDC) hourly integrated surface database (http://www.ncdc.noaa.gov/data-access/) for all of China. These sites are shown in Fig. 1. We

focus on 2 m temperature and relative humidity and 10 m wind speed and direction for model evaluation. Vertical profiles of meteorological variables were also obtained from the 325 m high observational tower located at the Institute of Atmospheric Physics (IAP), Chinese Academy of Sciences, Beijing ($39°58'28''$ N, $116°22'16''$ E). This provides independent measurements of temperature, relative humidity, wind speed and wind direction at 17 different height levels. Measurements of boundary layer mixing height were retrieved from aerosol lidar profiles at IAP (Yang et al., 2017), providing a valuable additional test

of model meteorological processes. Hourly concentrations of $NO_2$, CO, $SO_2$, $O_3$, $PM_{2.5}$ and $PM_{10}$ are available from the national monitoring network run by the China National Environmental Monitoring Center (CNEMC). In addition, over the October–November 2014 period detailed measurements of atmospheric pollutants and aerosol composition were made from the IAP tower. These include measurements of $NH_4$, $NO_3$, $SO_4$, and OC from an Aerodyne Aerosol Chemical Speciation Monitor (ACSM) instrument at 260 m altitude (Sun et al., 2016b) and from a High Resolution Aerosol Mass Spectrometer

(HR-AMS) instrument at the surface (Xu et al., 2015), and BC at the surface was measured with an Aethalometer. The size-segregated samples collected at the two heights were analyzed for water-soluble ions. Detailed procedures for the data analysis are described in Ng et al. (2011) and Sun et al. (2012).

## 3   Model Evaluation

To investigate the strengths and weaknesses of the model in representing air quality in China, the model was evaluated against

meteorological and pollutant measurements across all three domains and at the IAP tower site in Beijing.

### 3.1   Meteorology

We test the model performance with two sets of meteorological fields: Final Reanalysis data (FNL) from the National Centers for Environmental Prediction (NCEP) and ERA-Interim data from the European Centre for Medium-Range Weather Forecasts (ECMWF). Table 2 presents a domain-based comparison of the performance of simulated meteorological variables with

25 ground-based observations from the NCDC dataset when the model was run using these meteorological fields. With both sets of fields the average 2 m temperature is reproduced well over domains 2 and 3, but is slightly underpredicted over the largest domain, and is overpredicted for the single Beijing site. The Beijing observations are made at the airport on the outskirts of the city, and may not be representative of the wider region. The correlation coefficients are high over all three model domains (0.94–0.95) with both ECMWF and FNL fields. The surface relative humidity is underpredicted for all domains with both

sets of fields, although the biases are smaller and correlation coefficients higher with ECMWF data. The humidity is under-predicted by about 15% at the Beijing site and this may have implications for heterogeneous reactions and the hydroscopic growth of secondary aerosols. The 10 m wind speed is substantially underpredicted with both sets of fields for all domains,



**Table 2.** Comparison of observed and simulated meteorological variables using FNL and ECMWF fields

| | Number of | Obs. avg. | Sim avg | | Bias | | RMSE | | r | |
| | Stations | | FNL | ECMWF | FNL | ECMWF | FNL | ECMWF | FNL | ECMWF |
|---|---|---|---|---|---|---|---|---|---|---|
| 2-m Temperature (°C) | | | | | | | | | | |
| Beijing | 1 | 9.68 | 11.52 | 11.44 | 1.84 | 1.76 | 3.28 | 3.36 | 0.88 | 0.87 |
| D03 | 30 | 8.91 | 8.98 | 8.95 | 0.07 | 0.04 | 2.47 | 2.46 | 0.94 | 0.94 |
| D02 | 77 | 7.87 | 7.53 | 7.55 | -0.34 | -0.32 | 2.39 | 2.35 | 0.95 | 0.95 |
| D01 | 324 | 9.62 | 7.77 | 7.79 | -1.85 | -1.83 | 3.23 | 3.23 | 0.94 | 0.94 |
| 2-m Relative Humidity (%) | | | | | | | | | | |
| Beijing | 1 | 54.7 | 34.1 | 39.1 | -20.6 | -15.6 | 26.9 | 22.4 | 0.77 | 0.81 |
| D03 | 30 | 54.9 | 44.8 | 48.9 | -10.1 | -6.0 | 19.6 | 16.7 | 0.75 | 0.78 |
| D02 | 77 | 54.4 | 47.8 | 51.1 | -6.6 | -3.3 | 17.4 | 15.2 | 0.74 | 0.78 |
| D01 | 324 | 62.8 | 60.4 | 62.6 | -2.4 | -0.2 | 16.8 | 15.6 | 0.73 | 0.76 |
| 10-m Wind Speed (m s$^{-1}$) | | | | | | | | | | |
| Beijing | 1 | 5.41 | 2.27 | 2.24 | -3.14 | -3.17 | 4.98 | 5.09 | 0.72 | 0.69 |
| D03 | 30 | 5.73 | 3.26 | 3.20 | -2.47 | -2.53 | 4.60 | 4.65 | 0.62 | 0.61 |
| D02 | 77 | 6.18 | 3.60 | 3.55 | -2.58 | -2.63 | 4.52 | 4.55 | 0.67 | 0.66 |
| D01 | 324 | 5.67 | 3.38 | 3.36 | -2.29 | -2.31 | 4.29 | 4.30 | 0.60 | 0.61 |
| 10-m Wind Direction (°) | | | | | | | | | | |
| Beijing | 1 | 197.5 | 214.2 | 191.0 | 16.7 | -6.6 | 73.9 | 73.9 | 0.79 | 0.80 |
| D03 | 30 | 215.1 | 210.0 | 206.5 | -6.9 | -8.6 | 62.7 | 63.4 | 0.78 | 0.78 |
| D02 | 77 | 214.4 | 212.2 | 208.9 | -2.8 | -5.5 | 65.4 | 65.4 | 0.76 | 0.76 |
| D01 | 324 | 206.5 | 193.4 | 188.4 | -13.1 | -18.0 | 71.9 | 72.2 | 0.74 | 0.74 |

Hourly values are used for each station from 12 October to 19 November 2014. Where observation data are missing, model values were removed to ensure that sampling was consistent.

and performance is least good for the Beijing site where the bias is greater than 50% with both fields. However, the correlation coefficient at this site is slightly better than over the three model domains suggesting that the hourly variability in wind speeds is captured moderately well. If the underprediction of wind speeds extends above the surface, then this may lead to the build-up of gas-phase and aerosol species in the simulations and to overproduction of secondary aerosols due to unrealistic stagnation. The model captures the 10 m wind direction reasonably well with both sets of data and the correlation coefficient is close to 0.80 for Beijing. It is notable that the correlation coefficient improves for most variables between domain 1 and domain 3, as the model resolution increases from 27 km to 3 km. Based on these comparisons with meteorological observations, and on subsequent comparison of pollutant concentrations, we find that the model performs marginally better using the ECMWF meteorological fields. With these fields the model captures the timing of pollution episodes better, leading to more realistic pollutant behaviour, and we have therefore chosen ECMWF fields over FNL fields for our model studies.





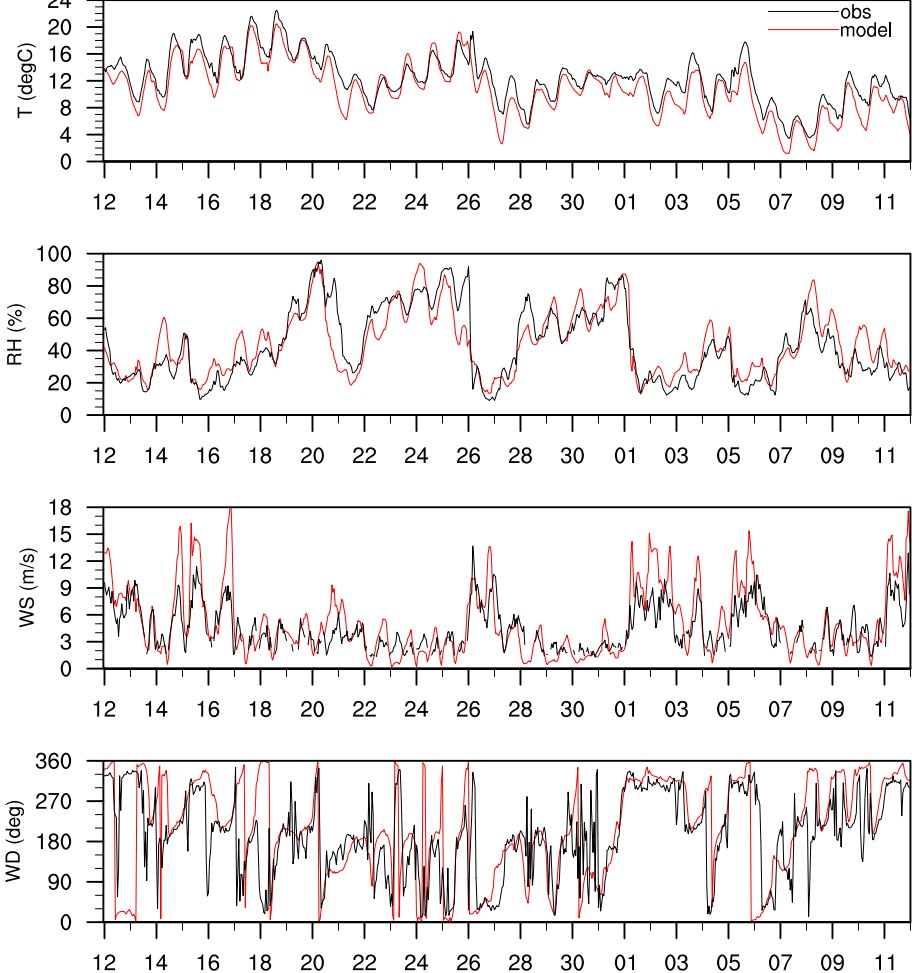

**Figure 2.** Comparison of meteorological measurements at 190–310 m on the IAP tower in Beijing with model simulations using ECMWF meteorological fields between 12 October and 12 November 2014.

Figure 2 presents an evaluation of meteorological variables with measurements from the IAP tower. We evaluate the model against measurements at 190–310 m (model level 4) to minimize the effects of buildings surrounding the site, which are not adequately resolved in the model. The daily maxima and minima in temperature are reproduced reasonably well with a small underestimation that averages less than 2 °C. The diurnal variations and averages for relative humidity, wind speed and wind direction are also captured well. The mean bias in relative humidity is 0.9% and the large underprediction at the airport meteorological station evident in Table 2 is not seen here, suggesting that it may be a surface level feature or reflect the overestimation of temperature at that location. Over the height of the tower (5 model levels) the diurnal variation in humidity drops by more than a factor of two, very similar to the reduction seen in the observations. The wind speed is slightly overestimated during windier periods, with a mean bias of $0.54\,\mathrm{m\,s^{-1}}$ Again, this suggests that the underestimation of 10-m wind speeds at mete-





orological stations seen in Table 2 is a surface feature in the model, and does not represent a systematic bias throughout the boundary layer. The synoptic patterns in all four variables are captured very well, highlighting the quality of the ECMWF meteorological data, and there is only one occasion on 20–21 October when substantial deviations in temperature and humidity are evident.

## 3.2 Air Quality

We ran the model for 41 days from 10 October to 19 November 2014 using ECMWF meteorology. The first 40 hours were set aside as model spin-up. A comparison of modelled pollutants against measurements from the CNEMC network is presented for October in Table 3 and the mean spatial distribution of $PM_{2.5}$ during October is shown in Fig. 3. We do not include the November period here because emission controls were implemented across Beijing and surrounding provinces from the beginning of November. A full time-series of the comparison to observations is shown in Fig. 4

Table 3 shows a comparison of the key gas-phase and particulate species for all the surface pollutant stations from the observation network over the corresponding model domains. The model overpredicts average surface $PM_{2.5}$ slightly (5–18%) across all domains. The correlation coefficient for hourly $PM_{2.5}$ improves with resolution from 0.47 for domain 1 to 0.63 for domain 3 and 0.68 for the 12 Beijing sites. The model underestimates $PM_{10}$ across all domains, although the biases are relatively small over Beijing. This underestimation may be attributed to neglect of mineral dust sources in the model, which play a relatively small role over Beijing at this time of year. CO is significantly underestimated for domain 1 but the biases reduce with increasing resolution and are smallest for the Beijing sites. The underestimation of CO for coarser domains may reflect the heterogeneity of sources, although the consistency of this bias and the relatively high levels of observed CO suggest an underestimate of CO sources across much of China in the emissions inventory. A similar effect is seen for $NO_2$, which is underestimated by 45% over the outer model domain, but by a much smaller margin over Northern China, and averages only 8% over the Beijing sites. While this may reflect an underestimate in emissions, the improvement is partly due to better representation of the emissions distribution for this shorter-lived pollutant on a finer grid. $SO_2$ is underestimated by 13% over domain 1 but is overestimated over Beijing by a factor of three. This large overestimation for Beijing can be attributed to the recent rapid reduction in emissions in the region between 2010 and 2014 that are not represented in the 2010 inventory (Zheng et al., 2018). Ozone shows a contrasting trend, with an overestimate of 50% for domain 1 reducing to 5% for domain 3 and a 3% underestimate over Beijing. This may reflect the bias in $NO_2$ concentrations, and is likely to be heavily influenced by the urban characteristics of most of the air quality stations.

For most pollutants, the correlation coefficient and slope improve substantially with resolution, and are better on a daily mean basis than at hourly resolution. This suggests that the day to day variability driven largely by regional meteorological processes is captured better than the diurnal variations driven by chemistry and local boundary layer mixing, as expected. This is particularly noticeable for ozone, although concentrations of this pollutant remain low at this time of year. Daily mean concentrations are typically used for most metrics of pollutant impacts on human health, and the reasonable model performance for daily averaged data suggests that it is suitable for assessment of these policy-relevant metrics.



**Table 3.** Comparison of pollutant concentrations with network measurements over the period 12–31 October 2014

| | Number of Stations | Obs | Sim | Bias | RMSE hourly/daily | r hourly/daily | slope hourly/daily |
|---|---|---|---|---|---|---|---|
| PM$_{2.5}$ ($\mu$g m$^{-3}$) | | | | | | | |
| Beijing stations | 12 | 108.3 | 126.2 | 17.9 | 86.7/66.7 | 0.68/0.78 | 0.83/0.93 |
| D03 | 137 | 92.6 | 109.3 | 16.7 | 72.2/52.2 | 0.63/0.74 | 0.71/0.80 |
| D02 | 375 | 75.8 | 87.9 | 12.1 | 63.9/48.6 | 0.60/0.69 | 0.65/0.71 |
| D01 | 1312 | 71.1 | 74.8 | 3.7 | 61.1/50.2 | 0.47/0.53 | 0.48/0.54 |
| PM$_{10}$ ($\mu$g m$^{-3}$) | | | | | | | |
| Beijing stations | 12 | 155.4 | 141.5 | -13.9 | 96.5/74.0 | 0.65/0.77 | 0.79/0.98 |
| D03 | 137 | 165.7 | 122.9 | -42.8 | 104.0/82.1 | 0.57/0.68 | 0.50/0.58 |
| D02 | 375 | 138.0 | 98.6 | -39.4 | 94.3/75.8 | 0.54/0.65 | 0.44/0.52 |
| D01 | 1312 | 121.0 | 82.2 | -38.8 | 89.0/76.7 | 0.42/0.47 | 0.32/0.37 |
| CO (ppm) | | | | | | | |
| Beijing stations | 12 | 1.11 | 0.94 | -0.17 | 0.63/0.43 | 0.60/0.75 | 0.46/0.61 |
| D03 | 137 | 1.17 | 0.83 | -0.34 | 0.87/0.72 | 0.29/0.34 | 0.21/0.22 |
| D02 | 375 | 1.14 | 0.66 | -0.48 | 0.88/0.79 | 0.33/0.37 | 0.20/0.20 |
| D01 | 1312 | 1.00 | 0.50 | -0.50 | 0.79/0.73 | 0.32/0.34 | 0.13/0.14 |
| NO$_2$ (ppb) | | | | | | | |
| Beijing stations | 12 | 39.09 | 36.09 | -3.00 | 19.33/11.10 | 0.62/0.80 | 0.66/0.83 |
| D03 | 137 | 29.75 | 25.88 | -3.87 | 18.95/14.32 | 0.47/0.54 | 0.45/0.51 |
| D02 | 375 | 24.86 | 19.45 | -5.41 | 16.99/13.21 | 0.49/0.55 | 0.44/0.50 |
| D01 | 1312 | 22.73 | 12.45 | -10.28 | 18.33/15.44 | 0.42/0.47 | 0.30/0.36 |
| SO$_2$ (ppb) | | | | | | | |
| Beijing stations | 12 | 3.92 | 12.27 | 8.35 | 11.88/10.55 | 0.27/0.52 | 0.68/1.74 |
| D03 | 137 | 13.28 | 14.47 | 1.19 | 13.66/9.66 | 0.21/0.31 | 0.22/0.24 |
| D02 | 375 | 12.23 | 13.21 | 0.98 | 13.19/9.01 | 0.24/0.34 | 0.26/0.28 |
| D01 | 1312 | 10.27 | 8.93 | -1.34 | 11.17/8.54 | 0.19/0.28 | 0.18/0.24 |
| O$_3$ (ppb) | | | | | | | |
| Beijing stations | 12 | 12.53 | 12.19 | -0.34 | 13.92/6.49 | 0.47/0.67 | 0.44/0.82 |
| D03 | 137 | 17.76 | 18.75 | 0.99 | 15.96/10.88 | 0.45/0.49 | 0.43/0.50 |
| D02 | 375 | 21.23 | 23.08 | 1.85 | 17.19/12.80 | 0.42/0.43 | 0.37/0.40 |
| D01 | 1312 | 21.44 | 32.29 | 10.85 | 22.44/17.03 | 0.29/0.27 | 0.27/0.25 |

The spatial distribution of mean PM$_{2.5}$ concentrations over 12–31 October is shown in Fig. 3. The distribution is captured reasonably well by the model, with the western parts of China showing clean air with concentrations less than 10 $\mu$g m$^{-3}$)





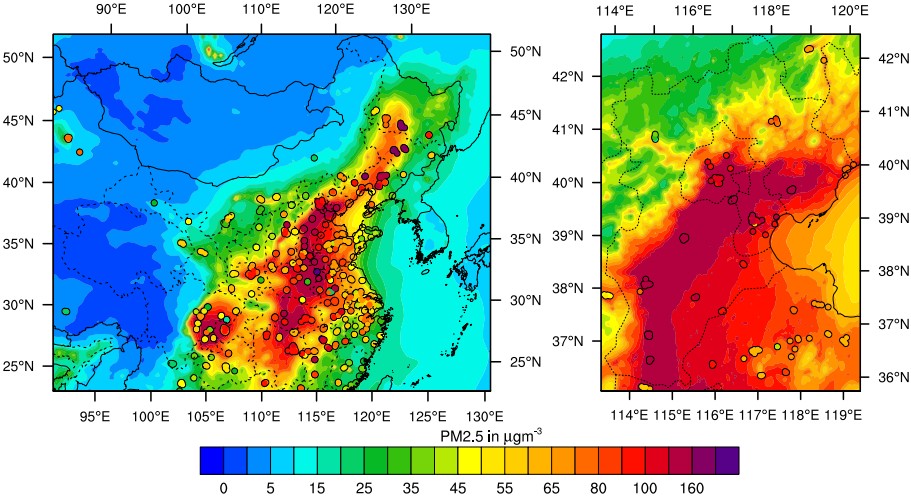

**Figure 3.** Average spatial distribution of $PM_{2.5}$ over the period 12–31 October 2014 for model domain 1 (left) and domain 3 (right) along with observations shown in circles.

while the eastern, more populous parts of the country show average concentrations of 70–150 $\mu g\,m^{-3}$. Key hot-spots over the North China Plain, Central China and the Sichuan Basin are reproduced, and concentrations in coastal regions are notably lower, matching observations. The North China Plain is one of the most densely populated parts of the country, incorporating major cities such as Beijing, Tianjin, and Shijiazhuang, and frequently experiences heavy haze episodes with high levels of

particulate matter (Wang et al., 2014; Gao et al., 2015a). Highest concentrations of $PM_{2.5}$ occur on the western side of the North China Plain, where they are trapped by southeasterly winds against the Taihang mountains, and this is reproduced well by the model. There is a notable east-west gradient as concentrations drop off eastwards towards the coast. Over the mountains to the northwest of Beijing concentrations are much lower, typically less than 40 $\mu g\,m^{-3}$.

     Figure 4 shows the time-series of key gas-phase and particulate pollutants averaged over the 12 network sites in Beijing.

The general synoptic and diurnal patterns of $PM_{2.5}$, $PM_{10}$, CO, $NO_2$ and $O_3$ are reproduced well by the model, including the magnitude of daily maxima and minima. $SO_2$ is greatly overestimated in October, reflecting recent rapid emission reductions in Beijing (Zheng et al., 2018), and this is consistent with the findings of previous studies (Chen et al., 2016a; Gao et al., 2015a; Guo et al., 2016). However, we note that $SO_2$ is reproduced much better from 15 November onwards, following the start of the heating season, highlighting the continuing major importance of this source. The observations show that the region experiences

clear synoptic patterns of pollutant build-up over 4–5 days followed by sudden clean-out which is typically associated with frontal passage from the northwest (Guo et al., 2014). These synoptic patterns are seen more clearly for particulate matter than for gas-phase pollutants like $NO_2$ and CO which exhibit a stronger diurnal signal reflecting chemical and dynamical processes. With the exception of $SO_2$, key pollutants and their variation over this period are reproduced well.

     A more critical test of model performance is made by comparison of aerosol composition with measurements at IAP over

this period, see Fig. 5. For all three episodes in October the model overestimates BC, $NO_3$ and $NH_4$ and underpredicts OC





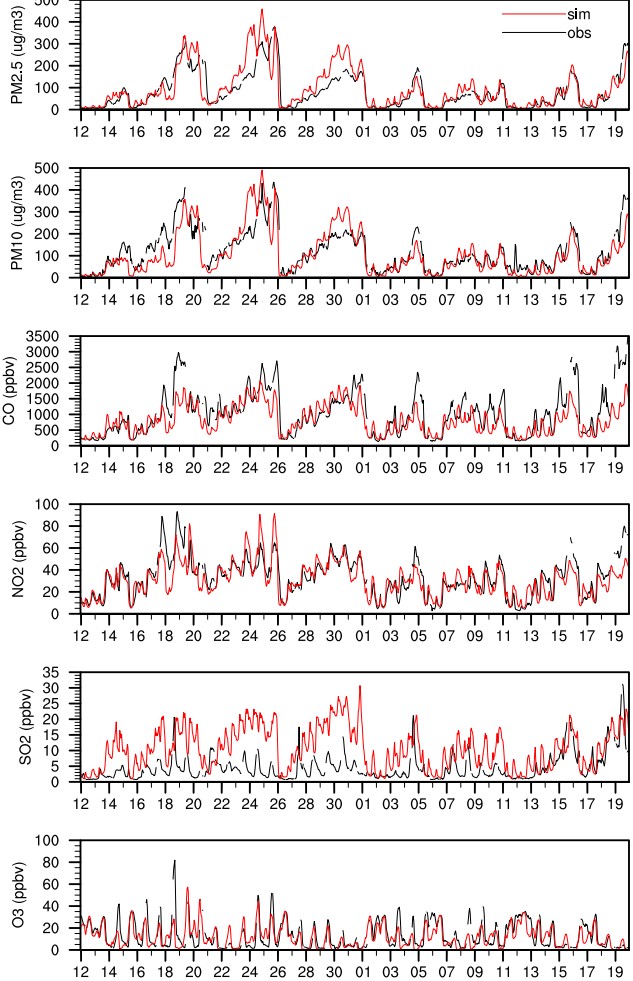

**Figure 4.** Mean time-series of surface pollutants over the 12 air quality stations in Beijing

and $SO_4$. The overestimation of BC likely reflects use of emissions for 2010, highlighting reductions between 2010 and 2014, but may also indicate insufficient removal in the model. The overprediction of $NO_3$ and $NH_4$ could be due to uncertainty in $NO_2$ and $NH_3$ emissions or to overestimated gas to particle conversion in the model. In particular, the model may overestimate secondary production of $NO_3$ and $NH_4$ during stagnant conditions such as those occurring during the three October episodes

5    (note the low wind speeds shown in Fig. 2), but matches better during the first half of November, when conditions are less stagnant. The underestimation of $SO_4$ occurs despite an overestimation of gas-phase $SO_2$, highlighting insufficient formation of $SO_4$ in the model. The underestimation of OC can be explained by the absence of secondary organic aerosol in the chemical mechanism we have used.

The relative proportions of $SO_4$ and $NH_4$ to other components are similar at the surface and 260 m, while the proportion

10    of OC is lower at 260 m and that of $NO_3$ is higher. The model concentrations of $NO_3$ and $SO_4$ are 10–15% and 30% lower





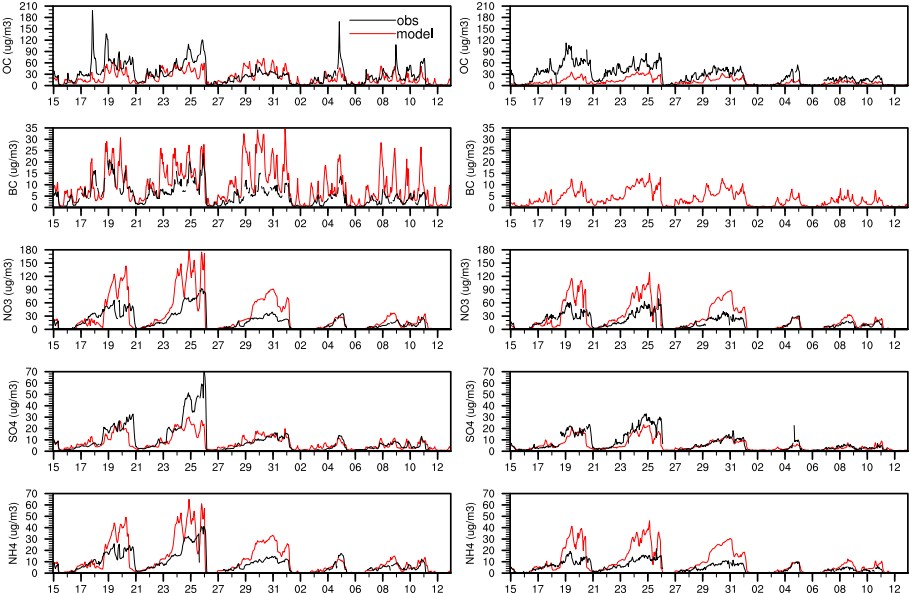

**Figure 5.** Measured and simulated aerosol components at the surface (left) and 260 m (right) on the IAP tower in Beijing

at 260 m, respectively, which is very similar to the observed reduction. The modelled $NH_4$ is 22% lower at 260 m, less than the observed reduction of 33%, and this may be attributed to an overproduction of secondary $NH_4$ in the model which reduces the vertical concentration gradient. Modelled OC falls off more quickly with altitude than in the observations (53% vs. 12%) and this is likely to be because the OC in the model is primary and therefore its vertical distribution reflects the surface source,
while there is more secondary OC in the observations (Sun et al., 2016b) which leads to a weaker gradient with altitude.

## 4   Investigating model sensitivity

While the baseline model simulation with ECMWF meteorological fields reproduces observed pollutant levels reasonably well, the comparisons have highlighted uncertainties associated with resolution, vertical mixing processes, and aerosol composition. We explore the sensitivity of our results to these factors here.

### 4.1   Model resolution

Running the model at high resolution comes at a substantial cost in computing resources and simulation time. To investigate the gains that increased horizontal resolution provides, we sample all three model domains at the 12 Beijing stations and compare the results with observations. We use two-way nesting, so results from the nested domains feed back to the parent domain. To eliminate this effect, we perform an additional simulation over the parent domain only. Table 4 shows a comparison with
measurements over Beijing in October for the different resolutions. In the nested simulation, $PM_{2.5}$ is overestimated by 14% for domain 1, 19% for domain 2 and 16% for domain 3, but is underestimated by 8% for the domain 1 simulation without





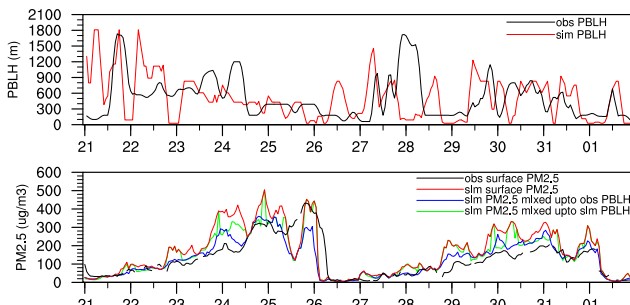

**Figure 6.** Simulated and observed boundary layer mixing height in metres (top) and simulated and observed $PM_{2.5}$ in $\mu g\,m^{-3}$ showing the effect of mixing up to the PBL height in the model (bottom) between 20 October and 2 November 2014.

nesting. Although the mean biases do not improve with higher resolution, reflecting the two-way nesting, there is a substantial improvement in the correlation coefficient (0.59 to 0.68) and slope (0.55 to 0.83) for $PM_{2.5}$ when nesting is used, and this occurs for other pollutants too. For many variables the results sampled at 9-km resolution (D02) are slightly better than those sampled at 3-km resolution (D03), although it should be noted that results at D02 are influenced by the higher-resolution

simulation at D03 through the two-way nesting. Results at 27-km resolution without nesting are substantially less good than those with two-way nesting, highlighting the important contribution of the coupling. We conclude that it is worth performing simulations at higher horizontal resolution as it gives a better representation of urban pollution levels.

## 4.2 Boundary layer mixing

Representing turbulent mixing processes in the boundary layer well is critical for simulating surface air quality. The nighttime
boundary layer under stable meteorological conditions is particularly difficult to model, and we find that the mixing height is often severely underpredicted (and is as low as 20 m on several occasions) causing pollutant concentrations to reach unrealistically high levels. Nudging meteorological fields to ECMWF reanalysis data reduces this bias but does not remove it. After testing a number of different boundary layer algorithms we selected the Yonsei University (YSU) scheme (Hong et al., 2006) as it provides the best overall match to lidar-derived observations of boundary layer height. However, stable conditions remain
a challenge for this scheme, and we therefore explore the sensitivity of simulated surface concentrations to boundary layer mixing under these conditions.

Figure 6 shows the time-series of simulated and observed planetary boundary layer (PBL) height. The observed PBL height was derived from lidar data at IAP using the cubic root gradient method of Yang et al. (2017). The simulated PBL height was diagnosed using the maximum decrease in the modelled $PM_{2.5}$ profile to ensure a consistent definition. We compare the
observed PBL height with the simulated height at IAP, and use $PM_{2.5}$ measurements from the surface pollutant station at Aotizhongxin, the closest station to the IAP site (within 2 km) to assess the effect on $PM_{2.5}$ concentrations. The PBL height shows highly variable behaviour over the day and from day to day. While the model average PBL height (514 m) is similar to the observed average height (509 m) over the haze episodes shown, the model severely underpredicts the nighttime PBL





**Table 4.** Impacts of model resolution on simulation of hourly pollutant concentrations in Beijing over 12–31 October 2014

|  | N points | Obs mean | Sim mean | Mean Bias | RMSE | r | slope |
|---|---|---|---|---|---|---|---|
| PM$_{2.5}$ ($\mu$g m$^{-3}$) |  |  |  |  |  |  |  |
| D03 (3-km) | 3171 | 108.4 | 126.2 | 17.8 | 86.7 | 0.68 | 0.83 |
| D02 (9-km) | 3171 | 108.4 | 128.7 | 20.3 | 87.4 | 0.69 | 0.85 |
| D01 (27-km) | 3171 | 108.4 | 123.1 | 14.7 | 86.1 | 0.68 | 0.81 |
| D01 (no nest) | 3171 | 108.4 | 99.2 | -9.2 | 83.2 | 0.59 | 0.55 |
| PM$_{10}$ ($\mu$g m$^{-3}$) |  |  |  |  |  |  |  |
| D03 | 2670 | 155.4 | 141.5 | -13.9 | 96.5 | 0.65 | 0.79 |
| D02 | 2670 | 155.4 | 143.6 | -11.8 | 96.6 | 0.65 | 0.80 |
| D01 | 2670 | 155.4 | 137.9 | -17.5 | 96.6 | 0.65 | 0.79 |
| D01 (no nest) | 2670 | 155.4 | 111.2 | -44.2 | 99.8 | 0.58 | 0.54 |
| CO (ppm) |  |  |  |  |  |  |  |
| D03 | 3074 | 1.11 | 0.94 | -0.17 | 0.63 | 0.60 | 0.46 |
| D02 | 3074 | 1.11 | 0.95 | -0.16 | 0.61 | 0.61 | 0.47 |
| D01 | 3074 | 1.11 | 0.88 | -0.23 | 0.61 | 0.64 | 0.44 |
| D01 (no nest) | 3074 | 1.11 | 0.68 | -0.43 | 0.73 | 0.62 | 0.31 |
| NO$_2$ (ppb) |  |  |  |  |  |  |  |
| D03 | 3080 | 39.09 | 36.09 | -3.00 | 19.33 | 0.62 | 0.66 |
| D02 | 3080 | 39.09 | 35.55 | -3.54 | 19.34 | 0.62 | 0.64 |
| D01 | 3080 | 39.09 | 31.92 | -7.17 | 18.33 | 0.67 | 0.62 |
| D01 (no nest) | 3080 | 39.09 | 21.81 | -17.28 | 24.74 | 0.60 | 0.48 |
| SO$_2$ (ppb) |  |  |  |  |  |  |  |
| D03 | 3074 | 3.92 | 12.27 | 8.35 | 11.88 | 0.27 | 0.68 |
| D02 | 3074 | 3.92 | 12.15 | 8.23 | 11.64 | 0.27 | 0.66 |
| D01 | 3074 | 3.92 | 10.91 | 6.99 | 9.82 | 0.32 | 0.69 |
| D01 (no nest) | 3074 | 3.92 | 6.47 | 2.55 | 5.57 | 0.29 | 0.40 |
| O$_3$ (ppb) |  |  |  |  |  |  |  |
| D03 | 3046 | 12.56 | 12.19 | -0.37 | 13.92 | 0.47 | 0.44 |
| D02 | 3046 | 12.56 | 12.71 | 0.15 | 13.94 | 0.47 | 0.43 |
| D01 | 3046 | 12.56 | 14.96 | 2.40 | 13.53 | 0.49 | 0.44 |
| D01 (no nest) | 3046 | 12.56 | 17.59 | 5.03 | 14.08 | 0.51 | 0.45 |

height on a number of occasions. Assuming that the PBL height reflects the efficiency of mixing in the boundary layer, we expect the model to overpredict surface pollutant concentrations under these stable nighttime conditions, and this is seen in the time-series of PM$_{2.5}$ at Aotizhongxin shown in Fig. 6. To account for misrepresentation of local boundary layer mixing,





we show the modelled $PM_{2.5}$ vertically-averaged up to the simulated mixing height, to minimise the effect of underestimated mixing, and up to the observed mixing height, to provide a clearer comparison against $PM_{2.5}$ observations. Mixing to the observed PBL height gives a substantial improvement in $PM_{2.5}$ levels compared to observations, particularly for the episodes of 21–25 October and 27 October–1 November when the model significantly underestimates the PBL height. The simulated

mean surface $PM_{2.5}$ concentration during the period is reduced from 169 to $130\,\mu g\,m^{-3}$ (the observed mean is $129\,\mu g\,m^{-3}$) and the RMSE is reduced from 94 to $65\,\mu g\,m^{-3}$.

These results highlight that accurate reproduction of surface pollutant levels with the model is tightly linked to how well it can reproduce PBL mixing. We note that the PBL shows a steady decline in height over the pollution episode during 21–25 October, and $PM_{2.5}$ shows a consistent build-up over the same period. This provides some observational evidence for the radiative

feedback between aerosol concentrations and mixing height, and this appears to be captured relatively well by the model, as shown in previous studies (Gao et al., 2015b). To further improve simulation of surface pollutant concentrations, additional research is needed to accurately model PBL mixing processes in urban environments. Profiles of aerosol and meteorological variables from high-resolution lidar measurements provide an important aid to such investigations.

### 4.3   Regional $NH_3$ emissions

The aerosol components $NO_3$ and $NH_4$ are overestimated in these simulations, as shown in Fig. 5. These components are governed by secondary production from their gaseous precursors $NO_2$ and $NH_3$. Since the concentration of $NO_2$ is close to that observed, we perform a short sensitivity study over the pollution episode from 21–25 October with $NH_3$ emissions over the North China Plain reduced by 50% to investigate the effect on aerosol composition. We find that the reduction in $NH_3$ emissions not only reduces $NH_4$ concentrations but also $NO_3$ and, to a lesser degree, $SO_4$ concentrations, see Fig. 7. This is

likely to be because $NH_3$ is the limiting reactant in the formation of $NH_4NO_3$ that directly controls the concentration of both $NH_4$ and $NO_3$ aerosols (Gao et al., 2016b) and consequently a reduction in $NH_4NO_3$ may suppress the secondary formation of $SO_4$ due to reduced aerosol surface area on which it can form. These reductions in aerosol components reduce total $PM_{2.5}$ concentration by approximately 26% bringing it closer to observed concentrations at the IAP site, see Table 5. Ammonia emissions were reported to be 1574 kt/yr over the Beijing-Tianjin-Hebei region in 2010 (Zhou et al., 2015) while those in the

MEIC emissions inventory used here are only 540 kt/yr. Given that our $NH_3$ emissions are already low compared with other studies (Kang et al., 2016), we do not reduce them further in this study. However, we demonstrate that $PM_{2.5}$ concentrations during this period are highly sensitive to $NH_3$ emissions, consistent with the findings of other studies (Zhang et al., 2016), and highlight this issue for further investigation.

## 5   APEC Emission Controls

The Asia-Pacific Economic Cooperation (APEC) summit was held from 10–12 November 2014 in Beijing, and was the focus of short-term emission controls to ensure good air quality over the period. Emission controls were applied in Beijing and surrounding regions including Tianjin city, the provinces of Hebei, Shanxi and Shandong, and Inner Mongolia Autonomous





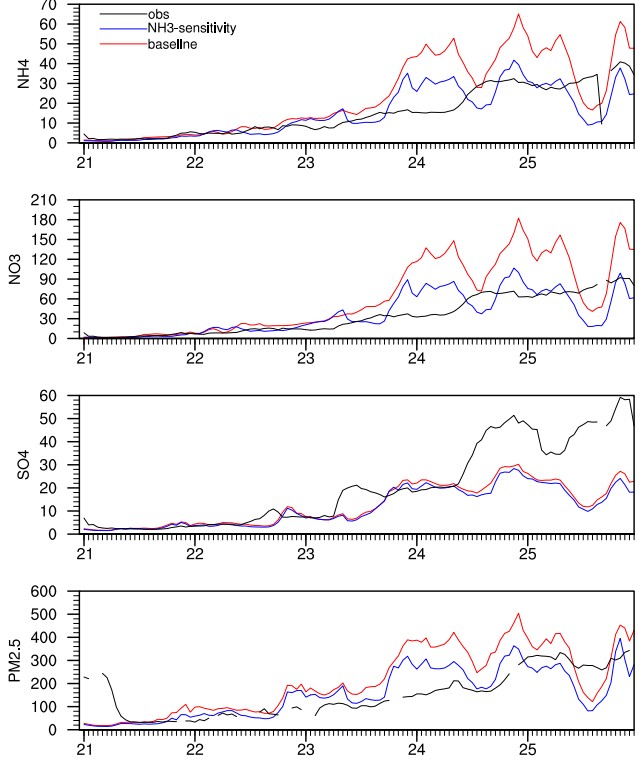

**Figure 7.** Time-series of aerosol components $NH_4$, $NO_3$ and $SO_4$ at the IAP site and $PM_{2.5}$ at Aotizhongxin showing simulated concentrations (in $\mu g\,m^{-3}$) from the baseline model run and reduced $NH_3$ emissions run compared to observations.

**Table 5.** Mean concentrations (in $\mu g\,m^{-3}$) at IAP during 21–25 October 2014

| Species | Control run | Reduced $NH_3$ run | Observations |
|---|---|---|---|
| $PM_{2.5}$ | 210.8 | 154.9 | 157.5 |
| $NO_3$ | 61.28 | 36.60 | 33.81 |
| $NH_4$ | 23.11 | 15.24 | 15.03 |
| $SO_4$ | 12.70 | 11.59 | 20.40 |

Region. More than 460 businesses with high emissions in Beijing were required to limit or stop their production during 3–12 November 2014 (Tang et al., 2015; Wang et al., 2016a; Guo et al., 2016). The number of private vehicles in operation over this period was reduced by about 50% through odd/even license-plate restrictions. Further, 9300 enterprises were suspended, 3900 enterprises were ordered to limit production, and more than 40,000 construction sites were shut down across the North

5 China region (Wang et al., 2016b; Tang et al., 2015). The start-up of municipal winter heating systems was delayed until



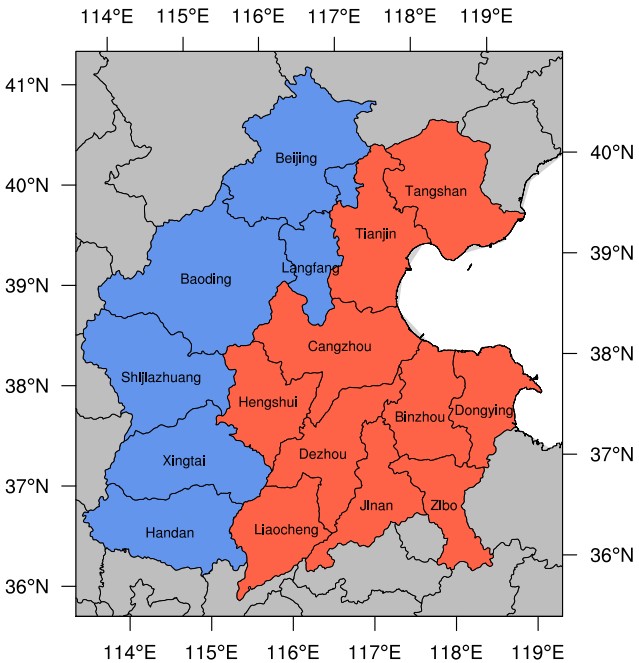

**Figure 8.** Map showing districts where major emissions controls were implemented during the APEC period. During phase 1 emissions were restricted in Beijing and western Hebei (blue) and in phase 2 controls were additionally applied over other parts of the North China Plain (red).

**Table 6.** Emission controls during APEC period

| Emission sector | Emission reduction (%) | |
| --- | --- | --- |
| | Beijing | Other Districts |
| Industry | 50 | 35 |
| Power | 50 | 35 |
| Agriculture | 40 | 30 |
| Residential | 40 | 30 |
| Transport | 40 | 30 |
| PM coarse (all sectors) | 80 | – |

APEC1: Beijing, Langfang, Baoding, Shijiazhuang, Xingtai, Handan
APEC2: APEC1 + Tangshan, Tianjin, Cangzhou, Hengshui, Dezhou,
Binzhou, Dongying, Zibo, Jinan and Liaocheng

15 November, after the summit. Implementation of these emission controls resulted in significant impacts on regional pollutant transport and local pollutant contributions (Meng et al., 2014; Sun et al., 2016b; Gao et al., 2017).





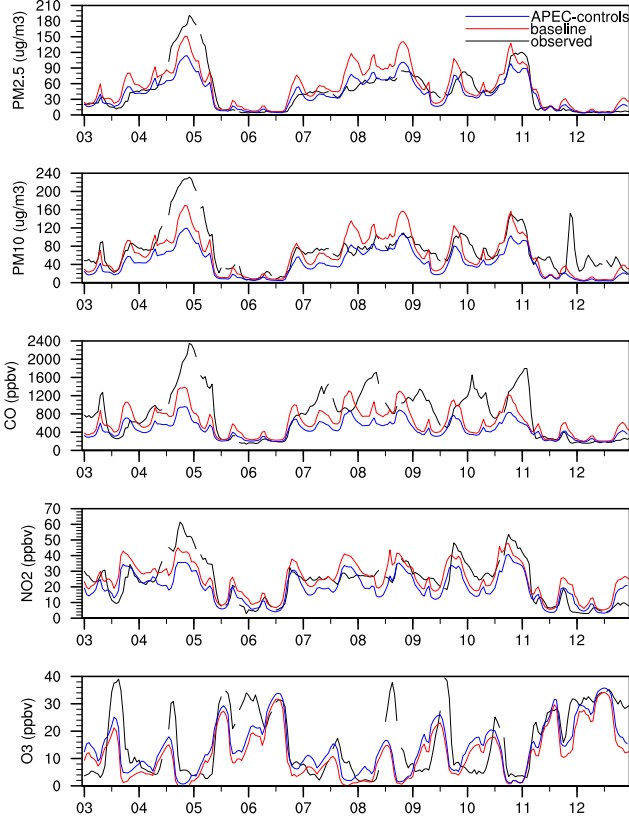

**Figure 9.** Time-series of surface pollutants averaged over the 12 measurement stations in Beijing during the APEC period.

Previous model studies of the APEC period have adopted different estimates of the emission reductions imposed (Guo et al., 2016; Gao et al., 2017; Wen et al., 2016; Liu et al., 2017; Wang et al., 2017). The most detailed study of emission reductions considered application of controls in two distinct phases (Wen et al., 2016), and we have chosen to implement these controls in our study, as the emission reductions applied are consistent with observation-based assessments of regional emission controls (Li et al., 2017b). During the initial phase (APEC1, 3–5 November), emission controls were implemented in Beijing and the western side of the North China Plain. In a subsequent phase (APEC2, 6–12 November) controls were applied over a wider region including eastern Hebei and parts of Shandong. We represent these controls in the model over the districts shown in Fig. 8, following Li et al. (2017b), and neglect smaller changes in emissions in other districts and more distant provinces. Controls were applied across different activity sectors following Wen et al. (2016) and Li et al. (2017b), see Table 6.

Figure 9 shows the effect of these controls on key pollutants over the period 3–12 November. There is a minor pollution episode over 4–5 November, and the model underestimates $PM_{2.5}$ levels over this period in the baseline run even without the emission controls. This may reflect an underestimation of OC as the simulation of secondary inorganic aerosol for these two days is relatively good (see Fig. 5). $PM_{2.5}$ levels are very well matched in the period 6–9 November leading up to the summit when applying the emission controls. $PM_{10}$ levels are underestimated in the simulations, but this is strongly influenced





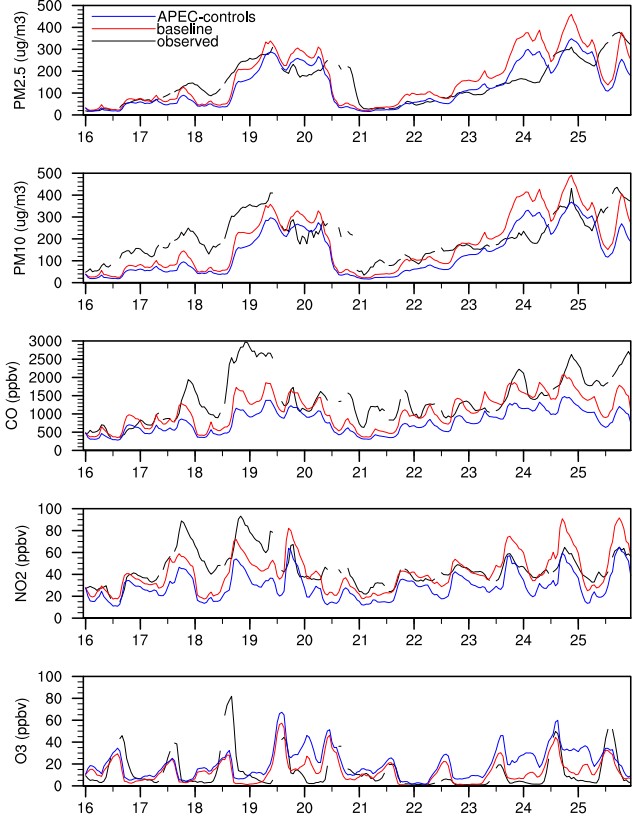

**Figure 10.** Time-series of surface pollutants averaged over the 12 measurement stations in Beijing during 16–25 October 2014.

by what may be a minor dust episode on 11–12 November, when coarse particles were high but $PM_{2.5}$ remained very low. Overall, the controls had a notable effect, reducing concentrations by 20–30% for all pollutants except $O_3$, which showed a small increase as expected for lower levels of NO. Over the critical 10–12 November meeting period, $PM_{2.5}$, $PM_{10}$, CO and $NO_2$ were reduced by 21%, 26%, 22% and 22% respectively, see Table 7. The reduction in $PM_{2.5}$ is very similar to the 22%

5 found in previous studies (Gao et al., 2017). However, the absolute improvement in air quality over the meeting period was small, averaging less than $10\,\mu g\,m^{-3}$ for $PM_{2.5}$, reflecting the relatively clean conditions over the period. Average $PM_{2.5}$ in the baseline simulation was $39\,\mu g\,m^{-3}$, close to the observed $36\,\mu g\,m^{-3}$. Under these conditions the key air quality standard, a 24-hour averaged $PM_{2.5}$ of $75\,\mu g\,m^{-3}$, corresponding to a Chinese Air Quality Index (AQI) of 100, would have been met in the model simulation even without the controls.

10   To explore the importance of meteorological conditions in contributing to the favourable air quality during the APEC period, we apply the same magnitude, location and duration of emission controls to the major pollution episode at the end of October. Fig 10 shows the effect of these controls on key pollutants over 16–25 October. The controls reduced pollutant concentrations by a larger amount than during the APEC period, but the relative improvements of 23–38% were very similar. The absolute pollutant concentrations were much higher than in November. This can be attributed to lower wind speeds and to winds



from the South and East bringing air from across the North China Plain, in contrast to the APEC period which experienced higher wind speeds and air from the clean northwest sector. The 3-day baseline average concentrations over 23–25 October for $PM_{2.5}$, $PM_{10}$, CO and $NO_2$ were $279\,\mu g\,m^{-3}$, $310\,\mu g\,m^{-3}$, 1.48 ppm and 53 ppb respectively, substantially exceeding air quality standards. The difference in baseline $PM_{2.5}$ concentrations between the October and November periods without

emission controls, 279 vs. $39\,\mu g\,m^{-3}$, highlights the dominant role played by meteorology in bringing clean air during APEC. The emission controls have a much larger absolute effect during the October episode than in the APEC period, with reductions in $PM_{2.5}$ of $65\,\mu g\,m^{-3}$ for 23–25 October, bringing average $PM_{2.5}$ levels down to $214\,\mu g\,m^{-3}$. However, this is insufficient to meet the standards needed for clean air of $75\,\mu g\,m^{-3}$. This indicates that the emission control policies applied would have failed to produce the desired results if the meeting had been held at the end of October.

Table 8 presents the effect on aerosol components and gas-phase pollutants at the IAP tower. During the emission controls in both the polluted October and cleaner November periods, primary components were reduced by 31–34% while secondary components were reduced by only 3–18%. This suggests that pollution episodes dominated by primary aerosols may be more easily controlled. This has serious implications for winter haze episodes over the North China Plain because much of the increase in aerosol loading is contributed by regional secondary aerosols (see Sun et al., 2016b).

To investigate the feasibility of meeting air quality standards during pollution episodes such as that on 21–25 October, we ran the model with all anthropogenic emissions removed over the North China Plain region shown in Fig. 8 from 16–25 October. The 3-day average concentrations over 23–25 October showed substantial reductions: 83% for $PM_{2.5}$, 82% for $PM_{10}$, 79% for CO, 99% for $NO_2$ and 88% for $SO_2$. Average $PM_{2.5}$ concentrations were reduced from $279\,\mu g\,m^{-3}$ to $48\,\mu g\,m^{-3}$, demonstrating that air quality standards can be met on highly polluted days, at least in theory, under the most stringent emis-

sion controls. From this simulation, and accounting for nonlinearity in secondary aerosol formation, we estimate that a 92% emission reduction over the 10 day period would have been needed to keep the average concentrations for 23–25th October below $75\,\mu g\,m^{-3}$. Even accounting for the model overestimation of average $PM_{2.5}$ during this period, driven principally by a high bias on 24 October, we find that an 85% emission reduction would be required, substantially more than is feasible realistically. It is clear from this analysis that emissions controls would need to be applied over a much wider area over neighbouring

provinces if the air quality standards in Beijing were to be met.

Finally, we analyse the full simulation period (12 October–19 November) to investigate how many days would meet the "blue-sky" criteria of 24-hour average $PM_{2.5}$ concentrations less than $75\,\mu g\,m^{-3}$ with and without the controls that were applied. To remove any model bias we use the relative benefits of controls derived from the model simulations and apply these reductions to the daily observed $PM_{2.5}$ levels averaged over the 12 air quality stations in Beijing to generate a scenario

representing the effect of emission controls. For 3–12 November, when emission controls were actually in place, we use the observed $PM_{2.5}$ concentrations unaltered. We assume a reduction in $PM_{2.5}$ of 22% at other times, representing an average of the responses seen over the October and November periods discussed above. To evaluate the effect of differences between modelled and observed aerosol composition on this scaling, given that primary aerosols are reduced more efficiently than secondary aerosols (Table 8), we apply the modelled reductions for each component to the observed component concentrations

to find the total reduction in PM. This is found to be 22% for both October and November periods, suggesting that this scaling



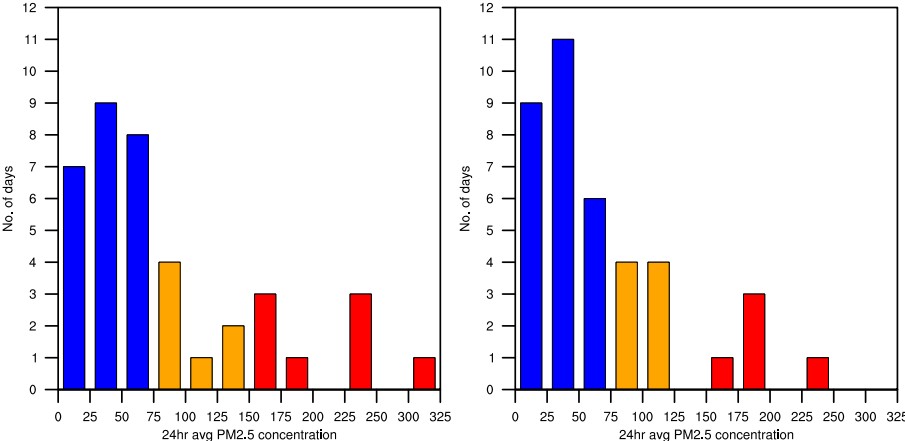

**Figure 11.** Frequency distribution of daily average $PM_{2.5}$ over 12 October–19 November 2014 showing the number of days meeting thresholds of $75\,\mu\mathrm{g\,m^{-3}}$ (blue) and $150\,\mu\mathrm{g\,m^{-3}}$ (blue plus orange) without (left panel) and with (right panel) emission controls.

is appropriate and robust to uncertainties in model aerosol composition. For the scenario with no controls, we apply an increase based on the November run of 16–33% for 3–12 November to estimate what the observations would have been, and use the observations on other days unaltered. With these scenarios we find that 15 of the 39 days considered failed to meet the blue-sky criteria of daily average $PM_{2.5}$ concentrations less than $75\,\mu\mathrm{g\,m^{-3}}$ without controls, and this fell to 13 days when the controls were implemented, a modest decrease of 2 days, see Fig. 11. However, if we choose a higher threshold of $150\,\mu\mathrm{g\,m^{-3}}$ (AQI of 200), the emission controls appear more effective, reducing the number of exceedances from 8 days to 5 days, and with a threshold of $200\,\mu\mathrm{g\,m^{-3}}$ (AQI of 250) the number of exceedances falls from 4 days to 1 day.

To organize a three-day meeting such as APEC successfully, all three days must individually meet the chosen air quality criteria. We find that without emission controls, only 9 out of 37 possible three-day time slots in our simulation period meet the criteria, including only 3 out of the 8 available during the APEC period of 3–12 November. Under the emission controls, the meeting could have been organized on 14 out of the 37 slots, including all 8 during early November. This suggests that the emission controls were only sufficient to provide an additional 5 time slots to hold a three-day event meeting the criteria. Interestingly, these all occur during the APEC period, highlighting that while favourable weather conditions were vital for meeting the air quality criteria, the emission controls provided critical support in achieving the $75\,\mu\mathrm{g\,m^{-3}}$ threshold needed to realise blue sky conditions. Specifically, in the absence of emission controls the first day of the APEC meeting (10 November) would have exceeded the air-quality standards. In this respect, it is reasonable to claim that the APEC emission controls were a success. However, it is clear that favourable meteorology was essential in making it possible for the emission controls to produce the marginal improvements needed to meet the air quality standards.

It should be noted that 23 out of the 37 possible three-day time periods, more than 60%, would not have met the standards even under the emission controls applied. It is therefore clear that much more stringent controls are needed in future to counter the effect of unfavourable meteorological conditions. While greater reductions in the magnitude of emissions are required, it





**Table 7.** Influence of emission controls averaged over Beijing air quality stations in October and November

| Species | Observed | | Model | |
|---|---|---|---|---|
| | Mean | Baseline | Controls | Improvement |
| APEC period (10–12 November) | | | | |
| $PM_{2.5}$ ($\mu g\,m^{-3}$) | 36.1 | 39.3 | 31.1 | 8.2 (20.9%) |
| $PM_{10}$ ($\mu g\,m^{-3}$) | 65.3 | 43.9 | 32.5 | 11.4 (26.0%) |
| CO (ppm) | 0.64 | 0.48 | 0.38 | 0.11 (22.0%) |
| $NO_2$ (ppb) | 19.0 | 20.6 | 16.0 | 4.6 (22.3%) |
| $SO_2$ (ppb) | 2.1 | 6.1 | 4.2 | 1.9 (30.8%) |
| $O_3$ (ppb) | 20.0 | 16.5 | 19.0 | -2.5 (-15.3%) |
| October period (23–25 October) | | | | |
| $PM_{2.5}$ ($\mu g\,m^{-3}$) | 216.1 | 278.8 | 213.7 | 65.1 (23.3%) |
| $PM_{10}$ ($\mu g\,m^{-3}$) | 263.8 | 309.6 | 236.4 | 73.2 (23.6%) |
| CO (ppm) | 1.77 | 1.48 | 1.05 | 0.44 (29.6%) |
| $NO_2$ (ppb) | 46.3 | 53.2 | 34.9 | 18.3 (34.4%) |
| $SO_2$ (ppb) | 4.0 | 18.6 | 11.6 | 7.0 (37.7%) |
| $O_3$ (ppb) | 11.4 | 15.2 | 26.7 | -11.5 (-75.5%) |

is important that these are applied over a much larger area, including in the neighbouring provinces that surround the North China Plain.

# 6  Conclusions

We have demonstrated that using a high-resolution nested air quality model we can reproduce the observed hourly variation
of major pollutants in Beijing during October–November 2014 reasonably well. We capture the synoptic drivers of air quality well, including the build-up of pollutants during pollution episodes and the subsequent cleaning effect of winds from the northwest. The concentrations of $PM_{2.5}$, the dominant pollutant in this season, are reproduced well, and we show that where the model is biased high, typically during nighttime, underlying weaknesses in the treatment of turbulent mixing in the planetary boundary layer are often responsible. We show that use of two-way nesting to high resolution brings a substantial benefit
in reproducing observed pollutant concentrations, even when comparing at the coarsest resolution used. Thorough evaluation against aerosol composition measurements over the period highlight some weaknesses in representation of key aerosol components, particularly the balance between $SO_4$, $NO_3$ and $NH_3$ which requires more detailed analysis.

We show that short-term emission controls played a valuable role in improving air quality over the APEC period, but that their overall contribution was relatively small, with average reductions of 20–26% for key pollutants. Without the controls,
average $PM_{2.5}$ levels are likely to have exceeded the national standard of $75\,\mu g\,m^{-3}$ on 10 November, the first day of the APEC



**Table 8.** Influence of emission controls at the IAP site in October and November

| Species | Observed | | Model | |
|---|---|---|---|---|
| | Mean | Baseline | Controls | Improvement |
| APEC period (10–12 November) | | | | |
| OC ($\mu$g m$^{-3}$) | 30.6 | 9.8 | 6.8 | 3.06 (31.1%) |
| BC ($\mu$g m$^{-3}$) | 3.4 | 4.8 | 3.2 | 1.63 (33.8%) |
| NO$_3$ ($\mu$g m$^{-3}$) | 10.9 | 8.6 | 8.3 | 0.27 (3.2%) |
| NH$_4$ ($\mu$g m$^{-3}$) | 5.0 | 3.8 | 3.5 | 0.30 (8.0%) |
| SO$_4$ ($\mu$g m$^{-3}$) | 4.8 | 3.5 | 2.9 | 0.60 (17.0%) |
| CO (ppm) | 2.60 | 0.68 | 0.52 | 0.16 (24.0%) |
| NO$_2$ (ppb) | 17.2 | 30.2 | 22.9 | 7.36 (24.3%) |
| SO$_2$ (ppb) | 10.4 | 9.4 | 6.3 | 3.07 (32.8%) |
| O$_3$ (ppb) | 3.5 | 17.6 | 21.6 | -4.04 (-23.0%) |
| October period (23–25 October) | | | | |
| OC ($\mu$g m$^{-3}$) | 60.5 | 39.5 | 26.7 | 12.79 (32.4%) |
| BC ($\mu$g m$^{-3}$) | 10.2 | 16.5 | 11.0 | 5.49 (33.2%) |
| NO$_3$ ($\mu$g m$^{-3}$) | 51.3 | 95.0 | 79.7 | 15.32 (16.1%) |
| NH$_4$ ($\mu$g m$^{-3}$) | 21.1 | 35.2 | 29.9 | 5.40 (15.3%) |
| SO$_4$ ($\mu$g m$^{-3}$) | 31.2 | 18.4 | 15.8 | 2.53 (13.8%) |
| CO (ppm) | 2.92 | 2.03 | 1.43 | 0.60 (29.5%) |
| NO$_2$ (ppb) | 44.2 | 78.1 | 57.8 | 20.30 (26.0%) |
| SO$_2$ (ppb) | 18.4 | 26.5 | 16.5 | 9.93 (37.5%) |
| O$_3$ (ppb) | 5.9 | 10.3 | 22.1 | -11.8 (-114.7%) |

meeting, but the effects were largely incremental, highlighting the important role played by favourable meteorology during the period. If the APEC meeting had been held at a different time, particularly at the end of October, air quality standards would not have been achieved with the emission controls applied. We find that the relative effect of the controls during the pollution episodes of late October are very similar to those during the clean APEC period, averaging 23% for PM$_{2.5}$. Much greater emission reductions of at least 85% would have been needed over the North China Plain region to bring pollutant levels down to meet air quality standards. It is clear that under the stable meteorological conditions present during these pollution episodes much more stringent emission controls are needed than those that were applied, and that these need to be implemented over a much wider region of Northern China. Our study demonstrates the value of short-term emission controls, but highlights that long-term, sustained emission reductions on a regional scale are required to bring blue skies to Beijing.





*Code and data availability.* The WRF-Chem code is available from http://www2.mmm.ucar.edu/wrf/users/download/. The namelist for the model and surface pollutant distributions generated in this study will be made available from the Lancaster University data archive at http://dx.doi.org/10.15125/XXXXXX.

*Author contributions.* TA, OW and ZW designed this study, and TA performed the model simulations and analysis. JL provided emissions
5  data and expertise on the model set-up, TY provided the lidar data and guidance on deriving PBL height, and YS, WX and ZW provided measurement data from the IAP tower. TA and OW prepared the manuscript with input from all coauthors.

*Competing interests.* The authors declare that they have no conflict of interest

*Acknowledgements.* OW thanks the UK Natural Environment Research Council for support under grants NE/N006925/1 and NE/N006976/1.
YS thanks the National Natural Science Foundation of China (Grant No. 91744207).

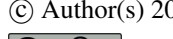



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
