# Peer review of "Effectiveness of short term air quality emission controls: A high-resolution model study of Beijing during the APEC period"

_Atmospheric Chemistry and Physics, 2018_

## Referee Comment (RC1) · Anonymous Referee #1 · 5 Feb 2019

Ansari et al. report numerical model simulations of air quality over China, focusing on Beijing, during the APEC summit in 2014. They investigate the benefit of short-term emission reduction measurements on near-surface pollutant concentrations and investigate uncertainty in model parameters. They conclude that choices of meteorological input data, model resolution and physical parameterisations are central to model performance, that emission controls were valuable in reducing pollutant levels but resulted in meeting air quality standards only because of favourable meteorological conditions.

General comments:

The authors present their research and results in a reasonably well-written manner,

and I could follow their reasoning with ease. There are, however a number of major concerns that I like to see addressed before this is published.

1) Their manuscript is too long, especially the sections on evaluating meteorological input datasets can surely be shortened and superfluous text, tables and figures moved to the supplementary material. This will improve readability and avoid loosing readers before the interesting stuff happens.

Particulate matter pollution is a intricate combination of source and sink processes which are individual for each chemical component, and their reaction to emission changes is as well. Hence all major components need to be represented (well) by the model to make believable predictions. The lack of secondary organic aerosol (up to 25% mass) and dust aerosol, as well as a strong overestimation of NO3(-) and underestimation of SO4(2-) are serious, yet total PM 2.5 mass miraculously works well. This can only be for the wrong reasons, which then has strong implications for the reliability of the results of sensitivity studies. The modelling system is in parts outdated and incomplete for an investigation of air quality in such a complex context. Hence:

2) Missing model components (SOA, dust) are readily available, especially for the WRF-Chem model used here, so they should be used

3) The SO2(g) to SO4(2-)(p) seems to be wrong and needs to be fixed

4) The emission inventory needs updating, and it should be done in a consistent manner rather than reducing SO2 by 60% and not touching the rest.

Detailed comments:

P1L19: this relationship is considered to be non-linear (e.g. Apte et al., 2015, Conibear et al., 2018a) according to recent findings - especially for high PM, benefits are much smaller. I suggest avoiding to give an exact number if this is merely the tangent at an (arbitrary?) point of a (now known-to-be) non-linear relation.

P2l5: comma missing after "Independent observational (. . .), modeling"

P2L20: this is not a thorough evaluation of met conditions

P3L1: this sounds like an arbitrary selection of processes to investigate - reason?

P3L15: This contradicts the manuscript by Sun et al. (2016b), cited here as reference. They state in their 'Implications' section: 'We demonstrate the response of aerosol composition, size distributions, and source contributions in Beijing to emission controls during APEC based on comprehensive measurements at both ground level and at a height of 260 m in urban Beijing. We observed large reductions of secondary aerosols during APEC, of 61–67% and 51–57% for SIA, and of 55% and 37% for SOA at 260 m and the ground site, respectively, whereas primary aerosols at ground level did not change in the same way. This large reduction of secondary aerosol is closely linked to the corresponding reduction of precursors over a regional scale, which suppresses the formation and growth of secondary aerosol by a factor of 2–3. Our results demonstrate that the achievement of "APEC Blue" is largely a result of significant reductions of secondary aerosol due to emission controls, although the mountain-valley breeze circulation also played a role.' (Sun et al., 2016b) How do you reconcile these seemingly contradictory statements? (Especially given that several of the authors of that publication are also co-authors here) Just because your "emphasis is largely on these components here" (p3l17) does not excuse missing the rest.

P3l30: only reducing SO2 emissions to account for the fact that the inventory is for 2010, whilst you are simulation 2014, is arbitrary - there are projections of Asian emissions available that allow to consistently project the whole dataset.

P3L33-34: again an indication that your model might be insufficient for the task at hand!

P4L2: MEGAN has been developed for North American conditions - can you be confident that it is applicable in China?

P5L20ff It is confusing to conflate comparisons over different spatial areas (domains D1, 2, 3) with different sets of observations given that you do 2-way nesting in your

model. In D1, meteorology and chemistry over the area covered by D2 are calculated on the D2 grid and then averaged back onto D1. Same for the region D3 in D2. So basically you are evaluating observations against model results where low pass filters of different strength were applied - no new information. At the same time you add new stations in the area not covered by inner domains. Weird.

P5L20ff Section 3.1 is overly detailed for a manuscript submitted to ACP, as it provides no further scientific insight beyond showing that meteorological variables can be simulated with good quality (known since 20 years), near-surface observations are difficult to match with a coarse grid (still quite some averaging to do at 3 km!) model, and that ECMWF IFS data seems to be a bit better than NCEP FNL (seemingly the case for 5+ years now). Hence I suggest: Shorten this paragraph to ∼10 lines, move Figure 2 and Table 2 and the rest of the paragraph into Supplementary material.

P8L12: so you do overestimate PM 2.5, but you don't have SOA and dust - what makes up for this missing component, so PM2.5 mass matches observations? Why? Does the replacement have the same formation pathways as SOA and dust? How can you pretend your model will react realistically to a change in precursor emissions given you are using different species (and formation pathways) to make PM 2.5 mass?

P8L20: This is roughly the reduction in NO2 expected from APEC emission cuts, no? So your error is roughly the magnitude of your signal, leading to quite a low signal-to-noise ratio that needs to be discussed.

P8L22: As you still have this large overestimation of SO2 in Beijing, how did you come up with the 60% reduction in emissions on P3L30? Why not more? There should be more up-to-date emission estimates for Beijing than MEIC 2010...

P8L25: O3 surface obs are notoriously difficult to interpret against model simulations due to the strong titration effects during the night, especially over urban areas. How does the maximum 8 hour O3 look like?

P8L27: by chance any ammonia measurements? HNO3?

P10L20ff: so here you go. PM2.5 and PM10 mass is right for the wrong reasons. Will a scenario simulation give the right answers, then?

P11L3: given that you underestimate SO3, NH3 will happily bind to NO3 to neutralize and form NH4NO3.

P11L6: Could technically be reasoned due to the fact that SO2 -> SO4— conversion takes some time, so most of your local SO4— might be imported. Given that you are underestimating SO2 outside of Beijing, this would make sense. But: it happens during stagnant conditions, so I would suggest that something seems seriously wrong with your model for seconary inorganic aerosols / SO2-SO4— conversion.

P12L3ff: see my previous comments on lacking model SOA.

P14Table4: a candidate to be put in the Supplemental Material

P15L1ff: it is unclear to me what you have done here - how could you mix modelled PM 2.5 up to simulated and observed PBL heights? Did you do additional simulations assimilating PBLH? Explain better!

P16L12: It should be made clear that 3 km average simulations over densely urbanized areas (think high-rise buildings) cannot realistically be expected to match an observation within that area due to the strong local topographical effects.

P15L21ff: SO4 is mainly formed through liquid-phase oxidation of SO2 in cloud droplets to H2SO4 and subsequent salt formation with NH3. Hence SO2->SO4— formation is typically not limited by aerosol surface area.

P15L21ff: NH3 preferably combines with SO4— to form (NH4)2SO4, only after most SO4— is depleted the remaining NH3 forms NH4NO3 (e.g. Seinfeld and Pandis, Atmospheric Chemistry and Physics, Wiley Interscience, 2012). Your SO4— is too low from the beginning, this "sensitivity study" hence does not take place in the right

chemical regime. How can you expect your results to be meaningful?

P15L30: We finally come to the topic of this manuscript. After 15 pages. This is too long. See my previous suggestions on how to reduce the extent of this work.

P19L1: I think it is an oversimplification that dust episodes only affect PM10, but not PM2.5. Apparently the APEC summit took place right in a slight dust episode, but you also do not simulate that component. WRF-Chem has multiple, easy to use dust schemes - why don't you just use them?

---

## Referee Comment (RC2) · Anonymous Referee #2 · 18 Feb 2019

Major comments:

1. The APEC emission control analysis (Section 5) is a bit confusing in terms of writing and additional modeling analysis is needed to support the authors' conclusion that meteorology played a more important role for good air quality during APEC. First, I suggest the authors put a summary at the beginning of the section to state their overall strategies to separate the relative role of emission control vs. meteorology. Second, to put this analysis in the context of previously published ones, I suggest the authors conduct a sensitivity run in which the emission reductions are implemented over the whole study period (Oct – Nov). The resulting changes in PM2.5 concentrations should be

compared to the 22% change the authors estimated. If the comparison is satisfactory, it can demonstrate the authors' simplified method is justified and such a method can be adopted by others.

2. On the evaluation of model meteorology (Section 3), I suggest the authors add a paragraph or two to state what meteorological factors/conditions are most different between the APEC and non-APEC period and to what extent the WRF model can reproduce such differences.

3. I concur with the first reviewer that the manuscript is too long and particularly the tables are tedious and do not add substantial values to the manuscript. I suggest Table 2-4 can be shortened (e.g. showing only the inner domain) and put the rest in the supplementary.

Minor comments:

1. The first line of the abstract: add "short-term" before emission controls.

2. Pg 3, line 15: the statement on little SOA response to emission changes is too assertive with only one reference as support. In fact, I don't agree with this statement because (1) emission controls can affect the biogenic-anthropogenic interactions (NOx-BVOC) which affect SOA and (2) there is considerable uncertainty surrounding the role of anthropogenic VOC emissions on SOA in China. Thus, I suggest the authors change the tone of the statement and acknowledge the uncertainty in their modeling exercise due to omitting of SOA.

3. Figure 2 and Figure 4: (1) label the APEC period; (2) add the month on the x-axis

4. Pg 9, line 9: the November period should be the October period.

5. All the time series figures should have the month on the x-axis.

Grammar:

1. Pg 2, line 6: add comma before modeling.

---

## Author Comment (AC1) · 7 May 2019

Response to reviewers' comments for "Effectiveness of short-term air quality emission controls: A high-resolution model study of Beijing during the APEC period" by Tabish Umar Ansari et al.

**Anonymous Referee #1**

Ansari et al. report numerical model simulations of air quality over China, focusing on Beijing, during the APEC summit in 2014. They investigate the benefit of short-term emission reduction measurements on nearsurface pollutant concentrations and investigate uncertainty in model parameters. They conclude that choices of meteorological input data, model resolution and physical parameterisations are central to model performance, that emission controls were valuable in reducing pollutant levels but resulted in meeting air quality standards only because of favourable meteorological conditions.

Thank you for providing useful and constructive comments on our manuscript. We address these points in turn below.

**General comments:**

The authors present their research and results in a reasonably well-written manner, and I could follow their reasoning with ease. There are, however a number of major concerns that I like to see addressed before this is published.

1) Their manuscript is too long, especially the sections on evaluating meteorological input datasets can surely be shortened and superfluous text, tables and figures moved to the supplementary material. This will improve readability and avoid loosing readers before the interesting stuff happens.

We appreciate this concern and have cut back the model evaluation section of the paper substantially, as suggested. Tables 2-4 have been significantly shortened and full versions have been moved to supplementary material along with additional figures. Further details of the sections shortened are included in response to specific comments below.

The modelling system is in parts outdated and incomplete for an investigation of air quality in such a complex context.

We chose to use the WRF-Chem model for this study as it is one of the very best available tools for air quality modelling at the scales considered here. We apply WRF-Chem version 3.7.1 as the most up-to-date at the time this study was started and adopted specific gas-phase and aerosol chemical mechanisms that have been well tested and evaluated for this region in previous published work (Gao et al., 2016a, 2016b; Guo et al., 2016; Chen et al., 2016). We acknowledge that the model has weaknesses, as does every other modelling tool, but we address these in the text, and argue that the model does not need to be perfect to provide useful and meaningful results for the conditions we explore here. Our concern for the model skill in representing the conditions in this period is clearly demonstrated by the extensive evaluation we have performed (which the reviewer notes, but asks us to cut back), and in our identification of key weakness that need to be addressed in future studies: the SO2-NO3-NH3 balance, SOA, and PBL mixing representation, none of which are yet completely understood let alone included reliably in the latest models. We hope that our studies guide future model development but contend that weaknesses in the model we have used do not substantially affect our conclusions, as our sensitivity studies have demonstrated.

Particulate matter pollution is a intricate combination of source and sink processes which are individual for each chemical component, and their reaction to emission changes is as well. Hence all major components need to be represented (well) by the model to make believable predictions. The lack of secondary organic aerosol (up to 25% mass) and dust aerosol, as well as a strong overestimation of NO3(-) and underestimation of SO4(2-) are serious, yet total PM 2.5 mass miraculously works well. This can only be for the wrong reasons, which then has strong implications for the reliability of the results of sensitivity studies.

We acknowledge that the treatment of particle composition remains imperfect, but argue that this does not undermine the conclusions we draw on wider emission controls. It is not necessary to represent all constituents perfectly to draw clear conclusions that emission reductions will reduce PM levels.

Chemical components are generally well-represented during the APEC (November) period (see table below) with the exception of OA which is underestimated. The biases are further reduced on correcting the components for boundary-layer mixing, and these results are now included for the full October-November period in Figure S1 and Table S5 in supplement.

|     | Model Avg. | Obs. Avg. |
|-----|------------|-----------|
| ос  | 10.12      | 25.17     |
| BC  | 4.81       | 3.12      |
| NO3 | 11.85      | 8.92      |
| NH4 | 4.73       | 4.35      |
| SO4 | 3.43       | 4.09      |

The overestimation of NO3 and underestimation of SO4 are relatively small in November, as shown above. Natural dust is not an important component of aerosol at this time of the year in Beijing, and anthropogenic sources of dust are already included. The total PM2.5 mass does not work well "miraculously"; it works reasonably well for November with an appropriate representation of composition excepting the underestimation of OA, and it doesn't work as well for October where there is an overestimation which we investigate and present in section 4. Total PM2.5 is overestimated in October principally due to insufficient boundary layer mixing as described in section 4.2 in the manuscript (see Figure 6, and we have included detail on the effect on aerosol components in the supplement, Figure S1).

**We have now added the following lines in the manuscript:**

P3L13: "Currently available SOA schemes are poorly parameterized for Chinese conditions and significantly underpredict SOA (Gao et al., 2016b, 2015b). SOA contributed to 17–23% of total ground-level fine particulate matter in Beijing for the October-November period investigated here, while secondary inorganic aerosols (SIA) contribute up to 62% by mass (Sun et al., 2016b). We consider the lack of SOA formation in the model in drawing our conclusions."

P19L13: "Since different primary and secondary aerosol components can respond differently to emission controls (Table 8), we use component-level percentage reductions from the model runs and apply them to the observed component concentrations to find the percentage reduction in total PM. This is found to be approximately 22% for both October and November periods based on the APEC-controls and October-controls runs suggesting that this scaling is appropriate and robust to uncertainties in model aerosol composition."

2) Hence: Missing model components (SOA, dust) are readily available, especially for the WRF-Chem model used here, so they should be used

Dust emissions can be included in WRF-Chem but are strongly sensitive to surface wind speeds and their variability (and hence fidelity in representing meteorological processes in dust source regions) and to settling processes. Dust representation in WRF-Chem is still an area of active development (LeGrand et al.,2019, GMD). However, natural dust is primarily a problem in Northern China in Spring time, and it is not a major contributor to PM levels during the October-November period examined here. We note also that anthropogenic primary PM2.5 emissions other than OC and BC are already included as passive dust in the model.

Formation of SOA is still relatively poorly understood and remains very challenging to represent fully in models, particularly under Chinese conditions (Gao et al., 2016b). WRF-Chem 3.7.1 has several options for representing SOA, most notably SORGAM and VBS methods. SORGAM is based on SOA formation via the absorptive partitioning of surrogate oxidation products of VOCs using SOA yields determined from smog chamber experiments and has been found to underestimate SOA by an order of magnitude in Beijing (Gao et al., 2016a). The VBS method represents multigenerational ageing of IVOCs/SVOCs but these processes need measurement constraints. Currently only 1D-VBS is available in WRF-Chem and this is not coupled with all gas-phase and aerosol mechanisms. It is not available with CBMZ-MOSAIC. It is available with SAPRC-MOSAIC but without aerosol direct and indirect effects (Zhang et al., 2015) and with MOZART-MOSAIC-4bin option which sacrifices details of aerosol growth processes (only two size bins are available to represent PM1). Even with an experimentally-constrained ageing framework built on 2D-VBS (not yet available in WRF-Chem), OA loadings are underestimated by 40% (medium yield scheme) at four long-term observational sites (Zhao et. al, 2016). Such underestimation appears to be common in most parts of China during different seasons, and is exacerbated during haze events (Chen et al., 2017). In light of these continuing uncertainties we have adopted the well-tested and relatively computationally efficient chemical mechanism CBMZ-MOSAIC which represents secondary inorganic aerosol formation along with primary organic aerosol.

**3) The SO2(g) to SO4(2-)(p) seems to be wrong and needs to be fixed**

We agree that conversion of  $SO_2(g)$  to  $SO_4^{2^2}(s)$  through known pathways (photochemistry and cloud chemistry) is inadequate in the model to explain huge mass yields in sulfate in North China Plain during winter, and this has been identified in previous WRF-Chem studies (Gao et al.,2016a, ACP, Chen et al.,2016, ACP). However, the actual formation pathway is still unknown. Chen et al.,2016, ACP implemented a RH-dependent pseudo first-order reaction for sulfate formation but were unable to capture the peaks during the pre-APEC period. Sulfate production during winter haze in China is still an open scientific question. Some particle-level hypotheses involving nitrogen chemistry in aerosol-water surface (Cheng et al., 2016, Science Advances) and others (Wang et al.,2016, PNAS) have been proposed but have not been parameterized for use in regional chemical transport models. To address this issue, we added additional primary sulfate in the model from the same sources as SO2 to compensate for these missing rapid reactions. This simple approach works well for the APEC period and two out of three episodes during October which were relatively drier, but it underestimates sulfate during the 21-25 October episode when the RH was high (see table S2 in supplement). To address the reviewers concerns we have discussed the implications of this assumption in Section 5 (Page 18 Line 7) of the paper.

4) The emission inventory needs updating, and it should be done in a consistent manner rather than reducing SO2 by 60% and not touching the rest.

Emission inventories always need updating, particularly over China where emissions are changing rapidly. However, this is not practical, particularly when working at 3 km resolution, and we have therefore made a compromise by adopting the most widely-used and evaluated emissions inventory (MEIC 2010) and adapting it to represent 2014 conditions with a simple scaling approach. MEIC 2014 has become available very recently but is not available at the 3 km grid resolution required here. We reduced SO2 emissions by 50% (not 60%) over the North China Plain in this study to reflect recent emission controls. This reduction was not arbitrary but based on the best information available and has been corroborated by recent studies. SO2 emission reduction over Eastern China between 2010-2015 has been estimated to be 48% through OMI satellite columns (Krotkov et al., 2016) and 45% through top-down emission estimates (Zheng et al., 2018). While there has been an increase in emissions for many species since 2010, recent clean air actions have reduced the emissions of key pollutants like NOx in 2014 to levels very similar to those in 2010 (Zheng et al., 2018). Therefore, our emission inventory provides a reasonably good representation of 2014 conditions and this is clear from our evaluation against observed pollutant levels.

**Detailed comments:**

P1L19: this relationship is considered to be non-linear (e.g. Apte et al., 2015, Conibear et al., 2018a) according to recent findings - especially for high PM, benefits are much smaller. I suggest avoiding to give an exact number if this is merely the tangent at an (arbitrary?) point of a (now known-to-be) non-linear relation.

This is a good point, and the statement has now been changed to: "It is estimated that outdoor air pollution, mostly by PM 2.5, leads to 3.3 million premature deaths per year worldwide, predominantly in Asia (Lelieveld et al., 2015)."

P2I5: comma missing after "Independent observational (...), modeling"

Now added

**P2L20: this is not a thorough evaluation of met conditions**

We agree but have not purported to do this; we merely point out that previous studies have largely neglected the role that meteorological processes play, and we aim to address this in our study. In response to reviewer 2, we have added a more detailed evaluation of meteorological conditions over the period in Tables S1 and S2.

**P3L1: this sounds like an arbitrary selection of processes to investigate - reason?**

We have changed P3L1 to "We present sensitivity studies to key physical and chemical processes in section 4". These processes were selected for evaluation after examining the simulation results, and the reasons for this are explained in the following section. We note at P12L5: "While the baseline model simulation with ECMWF meteorological fields reproduces observed pollutant levels reasonably well, the comparisons have highlighted uncertainties associated with resolution, vertical mixing processes, and aerosol composition. We explore the sensitivity of our results to these factors here."

P3L15: This contradicts the manuscript by Sun et al. (2016b), cited here as reference. They state in their 'Implications' section: 'We demonstrate the response of aerosol composition, size distributions, and source contributions in Beijing to emission controls during APEC based on comprehensive measurements at both ground level and at a height of 260m in urban Beijing. We observed large reductions of secondary aerosols during APEC, of 61–67% and 51–57% for SIA, and of 55% and 37% for SOA at 260m and the ground site, respectively, whereas primary aerosols at ground level did not change in the same way. This large reduction of secondary aerosol is closely linked to the corresponding reduction of precursors over a regional scale, which suppresses the formation and growth of secondary aerosol by a factor of 2–3. Our results demonstrate that the achievement of "APEC Blue" is largely a result of significant reductions of secondary aerosol due to emission controls, although the mountain-valley breeze circulation also played a role.' (Sun et al., 2016b) How do you reconcile these seemingly contradictory statements? (Especially given that several of the authors of that publication are also co-authors here) Just because your "emphasisis largely on these components here" (p3117) does not excuse missing the rest.

Sun et al.,2016 were first to report the observations made at the IAP tower and based their claim of secondary aerosol suppression due to emission controls on a simplified theoretical framework and a direct comparison of the two periods without assessment of the substantial meteorological differences between them. However, in this study we explore the same period in a meteorologically-resolved way with a regional atmospheric chemical transport model. It is clear that the reductions were largely due to meteorological changes (see table S2) and PM2.5 levels would not have been as high as during the pre-APEC period even without emission controls (figure 8). This is one of the key messages of our study and is highlighted in table 7 and in the conclusions at page 20 line 18. To address the reviewers concern we have changed the statement on page 3 to the following:

"SOA contributed to 17–23% of total ground-level fine particulate matter in Beijing for the October-November period investigated here, while secondary inorganic aerosols (SIA) contribute up to 62% by mass (Sun et al., 2016b)."

P3I30: only reducing SO2 emissions to account for the fact that the inventory is for 2010, whilst you are simulation 2014, is arbitrary - there are projections of Asian emissions available that allow to consistently project the whole dataset.

The reduction is based on best available evidence and is not arbitrary. Please see our response to point 4 above which cites relevant papers in support of our choice.

P3L33-34: again an indication that your model might be insufficient for the task at hand!

This fully addressed in our response to point 3 above.

P4L2: MEGAN has been developed for North American conditions - can you be confident that it is applicable in China?

This is indeed correct, and previous studies using MEGAN have pointed to inaccuracies in isoprene emissions in China (Situ et al., 2013, 2014; Han et al., 2013). However, biogenic emissions are relatively low in Beijing at this time of year and are more important for ozone than for PM2.5, and we are therefore confident that this bias does not affect the conclusions of our study.

P5L20 It is confusing to conflate comparisons over different spatial areas (domains D1, 2, 3) with different sets of observations given that you do 2-way nesting in your model. In D1, meteorology and chemistry over the area covered by D2 are calculated on the D2 grid and then averaged back onto D1. Same for the region D3 in D2. So basically you are evaluating observations against model results where low pass filters of different strength were applied - no new information. At the same time you add new stations in the area not covered by inner domains. Weird.

The aim of this section was to provide a broad overview of model performance over each geographical domain. We acknowledge that nesting influences results over the parent domain but note that the domains vary substantially in size and we respectfully point out that representation bias affects comparisons over different scales despite the nesting. However, we appreciate that our comparison could appear confusing, and have simplified Table 2 by focussing on the inner domain only. We separately investigate the benefits due to nesting by sampling Beijing stations only from all domains and we present this in a simplified version of Table 4.

P5L20 Section 3.1 is overly detailed for a manuscript submitted to ACP, as it provides no further scientific insight beyond showing that meteorological variables can be simulated with good quality (known since 20 years), near-surface observations are difficult to match with a coarse grid (still quite some averaging to do at 3 km!) model, and that ECMWF IFS data seems to be a bit better than NCEP FNL (seemingly the case for 5+ years now). Hence I suggest: Shorten this paragraph to ~10 lines, move Figure 2 and Table 2 and the rest of the paragraph into Supplementary material.

We have significantly shortened Section 3.1 as suggested, shortened Table 2 and moved the more detailed version into the supplementary material (Table S1). However, we have retained some discussion here so that the reader is clear on the depth of our evaluation. While it is known that the ECMWF data is a little better than NCEP FNL, this is not true everywhere, and it is important that we characterise this for the study location. It is also important to demonstrate how well meteorological features are captured at IAP and to show how they vary over the period to allow interpretation of the pollutant measurements available.

P8L12: so you do overestimate PM 2.5, but you don't have SOA and dust - what makes up for this missing component, so PM2.5 mass matches observations? Why? Does the replacement have the same formation

pathways as SOA and dust? How can you pretend your model will react realistically to a change in precursor emissions given you are using different species (and formation pathways) to make PM 2.5 mass?

 $PM_{2.5}$  is overestimated over certain periods due to insufficient PBL mixing (Figure 6, S1). We do not have component-level observations for  $PM_{1-2.5}$ . Within PM1, the extra SIA produced in the model compensates for SOA mass but only amounts to 17-23% of PM1 mass. The response to emission changes for up to 83% of PM1 mass is reliable as it is properly represented in the model. The reduction in remaining (17-23%) PM1 mass has some uncertainty (discussed in section 5) however, SOA and SIA often show simultaneous buildup and clean-up, and have similar size distributions as seen in measurements in Beijing for different periods (Zhao et al., 2017; Sun et al., 2016a,b).

P8L20: This is roughly the reduction in NO2 expected from APEC emission cuts, no? So your error is roughly the magnitude of your signal, leading to quite a low signal-to-noise ratio that needs to be discussed.

The bias in NO2 is small over Beijing, less than 10%, suggesting that the emissions are appropriate. The magnitude of the APEC emission cuts is much larger than this, so there is no problem with signal-to-noise ratio. The bias over the China domain referred to here is larger, but this reflects biases in other parts of China, outside the region affected by emission cuts, and includes the effect of representation biases; grid cells are large, while monitoring sites are principally in cities, so the model may underestimate short-lived pollutants.

P8L22: As you still have this large overestimation of SO2 in Beijing, how did you come up with the 60% reduction in emissions on P3L30? Why not more? There should be more up-to-date emission estimates for Beijing than MEIC 2010.

SO2 was reduced by 50%. Please see point 4 above for a detailed explanation.

P8L25: O3 surface obs are notoriously difficult to interpret against model simulations due to the strong titration effects during the night, especially over urban areas. How does the maximum 8 hour O3 look like?

Ozone is represented well during both daytime and nighttime as shown in Figure 4, suggesting that production and titration processes are both captured well. However, ozone is not a major pollutant at this time of year and therefore we do not devote further analysis to it in this paper.

P8L27: by chance any ammonia measurements? HNO3?

Unfortunately neither NH3 or HNO3 measurements were available during this period.

P10L20: so here you go. PM2.5 and PM10 mass is right for the wrong reasons. Will a scenario simulation give the right answers, then?

We do not have component level information for aerosol larger than PM1. Even within PM1, most of the underestimation of NO3, NH4 and BC happens during the October period and these components are much better captured during the November period. Results from scenario simulations are subject to uncertainties to physical and chemical processes investigated in section 4 but are tested for robustness in section 5 where we apportion component-level percentage reductions to observed component proportions.

Also, these component-level discrepancies are minor during the APEC (November) period (see response to point 1).

P11L3: given that you underestimate SO3, NH3 will happily bind to NO3 to neutralize and form NH4NO3.

Thank you for your comment. We have now added this reason in the manuscript at P11L13: "Some overestimation of NO3 can also be due to this underestimation of SO4 as sulfate decrease frees up ammonia to react with nitric acid and transfers it to the aerosol phase (Seinfeld and Pandis, 2006)." P11L6: Could technically be reasoned due to the fact that SO2 -> SO4 conversion takes some time, so most of your local SO4 might be imported. Given that you are underestimating SO2 outside of Beijing, this would make sense. But: it happens during stagnant conditions, so I would suggest that something seems seriously wrong with your model for secondary inorganic aerosols / SO2-SO4 conversion.

We address this in our response to point 3 above.

P12L3: see my previous comments on lacking model SOA.

We address this in our response to point 2 above.

P14Table4: a candidate to be put in the Supplemental Material

We have moved the full table to supplementary material as suggested but have retained PM2.5 comparisons to illustrate the importance of resolution for the major pollutants of interest here.

P15L1: it is unclear to me what you have done here - how could you mix modelled PM 2.5 up to simulated and observed PBL heights? Did you do additional simulations assimilating PBLH? Explain better!

At the model grid cell representing the IAP site we identified the model levels corresponding to the simulated and observed boundary layer height each hour. We vertically averaged the simulated PM2.5 up to these model levels to create two new time-series. This averaging was mass-weighted based on the thickness of each model level to conserve mass. We then compare these time-series with surface observations and simulated values at the surface to diagnose the effect of mixing on PM2.5 levels in the model. The analysis here involves post-processing and no additional simulations were performed. Figure 6 has been made clearer now and the legend has been updated. By "mixed" we mean vertical averaging up to the corresponding model level of the simulated or observed PBLH. We have also made this clear in the text now.

P16L12: It should be made clear that 3 km average simulations over densely urbanized areas (think high-rise buildings) cannot realistically be expected to match an observation within that area due to the strong local topographical effects.

Yes. We have acknowledged this on Page 6 Line 8 in section 3.1

P15L21: SO4 is mainly formed through liquid-phase oxidation of SO2 in cloud droplets to H2SO4 and subsequent salt formation with NH3. Hence SO2 $\rightarrow$ SO4 formation is typically not limited by aerosol surface area.

Thank you for your comment. We have now changed this sentence to: "However, reduction in SO4 concentrations is negligible (1  $\mu$ g m-3) because sulfate formation is only indirectly associated with NH3 availability (Tsimpidi et al., 2007)"

P15L21: NH3 preferably combines with SO4 to form (NH4)2SO4, only after most SO4 is depleted, the remaining NH3 forms NH4NO3 (e.g. Seinfeld and Pandis, Atmospheric Chemistry and Physics, Wiley Interscience, 2012). Your SO4 is too low from the beginning, this "sensitivity study" hence does not take place in the right chemical regime. How can you expect your results to be meaningful?

The aim of conducting this sensitivity study was to understand the response of NO3 and NH4 to ammonia emissions in the model to explore why it differs from the real world. We have added a sentence to explain this but have moved the figure into the supplement in response to the request to shorten this section.

P15L30: We finally come to the topic of this manuscript. After 15 pages. This is too long. See my previous suggestions on how to reduce the extent of this work.

We have substantially cut back on the earlier sections as detailed above..

P19L1: I think it is an oversimplification that dust episodes only affect PM10, but not PM2.5. Apparently the APEC summit took place right in a slight dust episode, but you also do not simulate that component. WRF-Chem has multiple, easy to use dust schemes - why don't you just use them?

We note that there is a brief spike in PM10 that occurs across all Beijing measurement stations on the evening of 11 November and which is not accompanied by a similar enhancement in PM2.5. In the absence of reliable evidence to the contrary, we attributed this observed feature to dust, and this remains the most likely source. However, we appreciate that it is difficult to explore this without more detailed evidence and have amended the text to downplay this as it does not have direct relevance to the conclusions of our study.

**Anonymous Referee #2 Major comments:**

1. The APEC emission control analysis (Section 5) is a bit confusing in terms of writing and additional modeling analysis is needed to support the authors' conclusion that meteorology played a more important role for good air quality during APEC. First, I suggest the authors put a summary at the beginning of the section to state their overall strategies to separate the relative role of emission control vs. meteorology. Second, to put this analysis in the context of previously published ones, I suggest the authors conduct a sensitivity run in which the emission reductions are implemented over the whole study period (Oct – Nov). The resulting changes in PM2.5 concentrations should be compared to the 22% change the authors estimated. If the comparison is satisfactory, it can demonstrate the authors' simplified method is justified and such a method can be adopted by others.

Thank you for these helpful suggestions. We have revised Section 5 as suggested to make the message clearer. We have already highlighted the role of meteorology in reducing pollutant levels in our statement "The difference in baseline PM 2.5 concentrations between the October and November periods without emission controls, 279 vs. 39  $\mu$ g m-3, highlights the dominant role played by meteorology in bringing clean air during APEC" (Page 17 line 11) but the revisions suggested have made this clearer.

In response to these comments, we have now conducted an additional 39-day sensitivity run with APEC2 controls implemented throughout the run. Considering all 39 days, we found a daily reduction of  $26\pm6\%$ , and a reduction of  $23\pm4\%$  for days with daily mean PM2.5 >75 µg m-3. This is very close to the 22% reduction that we used in our study which also accounted for the temporal application of emission controls (3 days of mild APEC1 controls followed by 7 days of more stringent APEC2 controls) and this gives us increased confidence in the approach we have taken. We have now updated the paragraph in section 5 that discusses the controls and no-controls scenario (Page 19 Line 29 to Page 20 Line 11) describing this new run and making other points clearer.

2. On the evaluation of model meteorology (Section 3), I suggest the authors add a paragraph or two to state what meteorological factors/conditions are most different between the APEC and non-APEC period and to what extent the WRF model can reproduce such differences.

Thank you for this valuable suggestion. We have added an analysis of the meteorological conditions before and during the APEC period and have presented this in table S2 in the supplement. We have added a few sentences at the end of section 3.1 to summarise this: "There are some marked differences in meteorological conditions between the non-APEC period (Episode 1: 15–20 Oct, Episode 2: 21–25 Oct, Episode 3: 26 Oct–1 Nov) and the APEC period (3–12 Nov). There is a seasonal temperature drop (7 °C) from October to November accompanied by a drop in relative humidity, increase in wind speed and a general. change in wind direction from SE to SW which are well captured by the model. For a more detailed meteorological comparison of the pre-APEC and APEC period see table S2 in supplement". We have not added more details as the paper is already long and because reviewer 1 indicated that this section needed to be cut back.

3. I concur with the first reviewer that the manuscript is too long and particularly the tables are tedious and do not add substantial values to the manuscript. I suggest Table 2-4 can be shortened (e.g. showing only the inner domain) and put the rest in the supplementary.

We appreciate this concern and have cut back the model evaluation section of the paper substantially, as suggested by both reviewers. Tables 2-4 have been significantly shortened as suggested and full versions have been moved to supplementary material.

Minor comments:

1. The first line of the abstract: add "short-term" before emission controls.

Now done.

2. Pg 3, line 15: the statement on little SOA response to emission changes is too assertive with only one reference as support. In fact, I don't agree with this statement because (1) emission controls can affect the biogenic-anthropogenic interactions (NOx-BVOC) which affect SOA and (2) there is considerable uncertainty surrounding the role of anthropogenic VOC emissions on SOA in China. Thus, I suggest the authors change the tone of the statement and acknowledge the uncertainty in their modeling exercise due to omitting of SOA.

We have changed the statement to: "Secondary organic aerosol (SOA) formation is not included in the chemical mechanism used here. Currently available SOA schemes are poorly parameterized for Chinese conditions and significantly underpredict SOA (Gao et al., 2016b, 2015b). SOA contributed to 17–23% of total ground-level fine particulate matter in Beijing for the October-November period investigated here, while secondary inorganic aerosols (SIA) contribute up to 62% by mass (Sun et al., 2016b). We consider the lack of SOA formation in the model in drawing our conclusions"

We have also included another sentence in Section 5 at Page 18 line 9: "percentage reduction in OC maybe overestimated because all OC is primary in the model"

3. Figure 2 and Figure 4: (1) label the APEC period; (2) add the month on the x-axis

Now done.

4. Pg 9, line 9: the November period should be the October period.

Do you mean Pg8, line9? Emission controls were applied during the November period and so we show results from October only in the figure.

5. All the time series figures should have the month on the x-axis.

Now done.

Grammar: 1. Pg 2, line 6: add comma before modeling.

Now done.

**References:**

Chen, Q., Fu, T.-M., Hu, J., Ying, Q., & Zhang, L. Modelling secondary organic aerosols in China. National Science Review, 4(6), 801–803. https://doi.org/10.1093/nsr/nwx129, 2017.

Cheng, Y., Zheng, G., Wei, C., Mu, Q., Zheng, B., Wang, Z., ... Su, H. Reactive nitrogen chemistry in aerosol water as a source of sulfate during haze events in China. Science Advances, 2(12), e1601530. https://doi.org/10.1126/sciadv.1601530, 2016

Chen, D., Liu, Z., Fast, J., and Ban, J.: Simulations of sulfate-nitrate-ammonium (SNA) aerosols during the extreme haze events over northern China in October 2014, Atmospheric Chemistry and Physics, 16, 10 707-10 724, https://doi.org/10.5194/acp-16-10707-2016, 2016

Gao, M., Carmichael, G. R., Wang, Y., Saide, P. E., Yu, M., Xin, J., Liu, Z., and Wang, Z.: Modeling study of the 2010 regional haze event in the North China Plain, Atmospheric Chemistry and Physics, 15, 22 781–22 822, https://doi.org/doi:10.5194/acp-16-1673-2016, 2015

Gao, M., Carmichael, G. R., Saide, P. E., Lu, Z., Yu, M., Streets, D. G., and Wang, Z.: Response of winter fine particulate matter concentrations to emission and meteorology changes in North China, Atmospheric Chemistry and Physics, 16, 11 837–11 851,https://doi.org/10.5194/acp-16-11837-2016, 2016

Guo, J., He, J., Liu, H., Miao, Y., Liu, H., and Zhai, P.: Impact of various emission control schemes on air quality using WRF-Chem during APEC China 2014, Atmospheric Environment, 140, 311–319, https://doi.org/10.1016/j.atmosenv.2016.05.046, 2016.

Han, K. M., Park, R. S., Kim, H. K., Woo, J. H., Kim, J., & Song, C. H. (2013). Uncertainty in biogenic isoprene emissions and its impacts on tropospheric chemistry in East Asia. Science of the Total Environment, 463–464, 754–771. https://doi.org/10.1016/j.scitotenv.2013.06.003

Krotkov, N. A., McLinden, C. A., Li, C., Lamsal, L. N., Celarier, E. A., Marchenko, S. V., Swartz, W. H., Bucsela, E. J., Joiner, J., Duncan, B. N., Boersma, K. F., Veefkind, J. P., Levelt, P. F., Fioletov, V. E., Dickerson, R. R., He, H., Lu, Z., and Streets, D. G.: Aura OMI observations of regional SO2 and NO2 pollution changes from 2005 to 2015, Atmos. Chem. Phys., 16,4605–4629, https://doi.org/10.5194/acp-16-4605-2016, 2016.

Legrand, S. L., Polashenski, C., Letcher, T. W., Creighton, G. A., Peckham, S. E., & Cetola, J. D. The AFWA dust emission scheme for the GOCART aerosol model in WRF-Chem v3 . 8 . 1. Geoscientific Model Development, 12, 131–166., 2019

Li, G., Bei, N., Cao, J., Huang, R., Wu, J., Feng, T., ... Molina, L. A Possible Pathway for Rapid Growth of Sulfate during Haze Days in China. Atmospheric Chemistry and Physics, 17, 3301–3316. https://doi.org/10.5194/acp-2016-994, 2017

Situ, S., Guenther, A., Wang, X., Jiang, X., Turnipseed, A., Wu, Z., ... Wang, X. (2013). Impacts of seasonal and regional variability in biogenic VOC emissions on surface ozone in the Pearl river delta region, China. Atmospheric Chemistry and Physics, 13(23), 11803–11817. https://doi.org/10.5194/acp-13-11803-2013

Situ, S., Wang, X., Guenther, A., Zhang, Y., Wang, X., Huang, M., ... Xiong, Z. (2014). Uncertainties of isoprene emissions in the MEGAN model estimated for a coniferous and broad-leaved mixed forest in Southern China. Atmospheric Environment, 98, 105–110. https://doi.org/10.1016/j.atmosenv.2014.08.023

Sun, Y., Wang, Z., Wild, O., Xu, W., Chen, C., Fu, P., ... Worsnop, D. R. "APEC Blue": Secondary Aerosol Reductions from Emission Controls in Beijing. Scientific Reports, 6(November 2015), 20668. https://doi.org/10.1038/srep20668, 2016a

Sun, Y., Chen, C., Zhang, Y., Xu, W., Zhou, L., Cheng, X., ... Wang, Z. Rapid formation and evolution of an extreme haze episode in Northern China during winter 2015. Sci Rep, 6(January), 27151. https://doi.org/10.1038/srep27151, 2016b

Wang, G., Zhang, R., Gomez, M. E., Yang, L., Levy Zamora, M., Hu, M., ... Molina, M. J. Persistent sulfate formation from London Fog to Chinese haze. Proceedings of the National Academy of Sciences of the United States of America, 113(48), 13630–13635. https://doi.org/10.1073/pnas.1616540113, 2016

Zhang, Y., Zhang, X., Wang, L., Zhang, Q., Duan, F., & He, K. Application of WRF/Chem over East Asia: Part I. Model evaluation and intercomparison with MM5/CMAQ. Atmospheric Environment, 124, 285–300. https://doi.org/10.1016/j.atmosenv.2015.07.022, 2015.

Zhao, J., Du, W., Zhang, Y., Wang, Q., Chen, C., Xu, W., ... Sun, Y. Insights into aerosol chemistry during the 2015 China victory day parade: results from simultaneous measurements at ground level and 260 m in Beijing. Atmospheric Chemistry and Physics, (March), 3215–3232. https://doi.org/10.5194/acp-2016-695, 2017

Zheng, B., Tong, D., Li, M., Liu, F., Hong, C., Geng, G., Li, H., Li, X., Peng, L., Qi, J., Yan, L., Zhang, Y., Zhao, H., Zheng, Y., He, K., and Zhang, Q.: Trends in China's anthropogenic emissions since 2010 as the consequence of clean air actions, Atmospheric Chemistry and Physics, 2018, 1–27, https://doi.org/10.5194/acp-2018-374, 2018.